# FADS1-arachidonic acid axis enhances arachidonic acid metabolism by altering intestinal microecology in colorectal cancer

Chunjie Xu[1,4], Lei Gu [1,4], Lipeng Hu [2,4], Chunhui Jiang[1,4], Qing Li[2], Longci Sun[1], Hong Zhou[1], Ye Liu[1], Hanbing Xue [3] ✉, Jun Li [2] ✉, Zhigang Zhang [2] ✉, Xueli Zhang [2] ✉ & Qing Xu [1] ✉

Colonocyte metabolism shapes the microbiome. Metabolites are the main mediators of information exchange between intestine and microbial communities. Arachidonic acid (AA) is an essential polyunsaturated fatty acid and its role in colorectal cancer (CRC) remains unexplored. In this study, we show that AA feeding promotes tumor growth in AOM/DSS and intestinal specific $Apc^{-/-}$ mice via modulating the intestinal microecology of increased gram-negative bacteria. Delta-5 desaturase (FADS1), a rate-limiting enzyme, is upregulated in CRC and effectively mediates AA synthesis. Functionally, FADS1 regulates CRC tumor growth via high AA microenvironment-induced enriched gram-negative microbes. Elimination of gram-negative microbe abolishes FADS1 effect. Mechanistically, gram-negative microbes activate TLR4/MYD88 pathway in CRC cells that contributes FADS1-AA axis to metabolize to prostaglandin E2 (PGE2). Cumulatively, we report a potential cancer-promoting mechanism of FADS1-AA axis in CRC that converts raising synthesized AA to PGE2 via modulating the intestinal microecology of gram-negative.

The intestine is responsible for food digestion, nutrient absorption, and other functions[1]. Carbohydrates, amino acids, lipids, and trace elements are all absorbed in the intestine[2]. The intestine is in physically dynamic balance, modulating the physiological functions of other organs to maintain regular functions. Dietary imbalance and environmental factors can severely disrupt the function of the intestinal epithelial and immune cells, induce inflammation, destroy the intestinal barrier, and ultimately lead to intestinal diseases[3]. High-fat diets severely impair the body's normal metabolic processes and alter the intestinal microenvironment and immune homeostasis[4]. The tumorigenesis of colorectal cancer (CRC) is associated to a large extent with the intake of diets rich in fat[5,6], one of the core factors for the high CRC incidence in western countries[7].

Fatty acids are crucial for energy metabolism[8]. Furthermore, unsaturated fatty acids are converted into important regulatory substances, such as prostaglandins (PGs) and leukotrienes, participating in the systematic regulation of physical activities. Arachidonic acid (AA) is a vital polyunsaturated fatty acid (PUFA) for cellular physiological processes, including inflammation and immunity[9]. AA is primarily derived from linoleic and linolenic acids in mammals' metabolism. Delta-5 desaturase (D5D) and delta-6 desaturase (D6D), encoded by FADS1 and FADS2, respectively, catalyze the controlling steps in AA synthesis[10]. Large-scale sequences based on East Asian populations show that FADS1 is an important risk factor in CRC[11]. However, the role and mechanism of AA and FADS1 in CRC are remains unknown.

[1]Department of Gastrointestinal Surgery, Renji Hospital, School of Medicine, Shanghai Jiao Tong University, Shanghai, China. [2]State Key Laboratory of Oncogenes and Related Genes, Shanghai Cancer Institute, Shanghai Jiao Tong University, Shanghai, China. [3]Division of Gastroenterology and Hepatology; Key Laboratory of Gastroenterology and Hepatology, Ministry of Health; Renji Hospital, School of Medicine, Shanghai Jiao Tong University, Shanghai, China. [4]These authors contributed equally: Chunjie Xu, Lei Gu, Lipeng Hu, Chunhui Jiang. ✉e-mail: medxue@126.com; junli@shsci.org; zzhang@shsci.org; xlzhang@shsci.org; renjixuqing@163.com

Gut microbial communities are known to be significantly altered in CRC patients. *Escherichia coli* (*Es.coli*), *Enterobacter faecalis* (*En.faecalis*), and *Fusobacterium nucleatum* are notably increased in CRC tissues. These changes in intestinal microecology are vital for tumor cell proliferation, metastasis, and chemoresistance[12–14]. However, how these specific gut microbes colonize and propagate in CRC tissue is not understood. Generally, colonocyte metabolism shapes the microbiome[15]. Metabolites might be the main mediators of information exchange between tumors and the microbiome. Known metabolites, such as butyrate and short-chain fatty acids, could regulate intestinal microbial communities[16].

Herein, we identify the FADS1-AA axis in CRC cells that modulate the intestinal microecology of enriched gram-negative bacteria via creating a high AA microenvironment. Enriched gram-negative microecology accelerated AA converting to prostaglandin E2 (PGE2) and eventually promotes CRC tumorigenesis.

## Results

### AA promotes tumor growth of CRC

Initially, after measuring AA levels, the results showed an increasing trend from the normal colon to colon adenoma to CRC tissue (Fig. 1a). We established an inflammation-induced CRC model by using the azoxymethane (AOM)/dextran sodium sulphate (DSS) method (Fig. 1b). AA feeding promoted tumor development as measured by larger tumor numbers, a higher tumor grade, and increased cell proliferation (Fig. 1c–e). Next, a spontaneous model of CRC was established by crossing *Apc* $^{flox/flox}$ and *Lgr5-EGFP-IRES-CreERT2* mice to generate intestine-specific *Apc*$^{-/-}$ mice, which spontaneously developed CRC in response to tamoxifen (Fig. 1g). Again, AA feeding showed a tumor-promoting role in the development of CRC (Fig. 1h-j). Moreover, AA feeding obviously increased the AA and downstream prostaglandins levels in tumor tissues, especially PGE2 (Fig. 1f, k).

### AA contributes to CRC process via regulating gut microbes

Interestingly, in vitro and subcutaneous tumor experiments showed no obvious effects of AA on CRC cell proliferation and CRC development, respectively (Fig. 2a, b and Supplementary Fig. 1a-c), suggesting the tumor promotion of AA in vivo may be dependent on the intestinal microenvironment. Therefore, we fed mice with a high AA diet (Fig. 2c) and measured response in microflora using 16s rRNA sequence. AA-induced prominent alteration of the composition of the gut microbes compared with the controls, and this alteration was noted without significant changes to total operational taxonomic unit (OTU). In detail, AA feeding significantly increased the proportion of gram-negative bacteria and decreased the proportion of gram-positive bacteria (Fig. 2d-f), which was consistent with the results obtained in case of *Apc* wild type (WT) mice (Supplementary Fig. 1d, e). Subsequently, in both AOM/DSS-indcued mice and intestine-specific *Apc*$^{-/-}$ mice, AA feeding again showed similar changes in the gut microbes (Fig. 2g and Supplementary Fig. 2a-c). Gut microbial elimination via quadruple-antibiotic administration completely abolished the tumor-promoting activity of AA (Fig. 2h-k and Supplementary Fig. 2d-i).We also collected stools from AA-fed and normal chow (NC)-fed mice for fecal transplantation and the results showed that feces from AA-fed mice (stool-AA) significantly promoted tumor growth compared with feces from NC-fed mice (stool-NC) (Fig. 2h-k and Supplementary Fig. 2f-i). Then, we conducted a selective antibiotic experiment. In detail, aztreonam was used to eliminate gram-negative bacteria and vancomycin to eliminate gram-positive bacteria. Removing gram-negative bacteria by aztreonam completely eliminated the tumor-promoting effect of AA and removing gram-positive bacteria by vancomycin had no effect on tumor-promoting role of AA (Fig. 2h-k and Supplementary Fig. 2f-i). Further analysis revealed that stool-AA and AA-fed+vancomycin groups both showed a similar gut microbiota composition with AA-fed groups, and stool-NC and AA-fed+aztreonam groups both

showed a similar gut microbiota composition with NC-fed groups (Supplementary Fig. 2j). These results indicated that AA promoted CRC process by creating a special intestinal microbiota environment of gram-negative.

### FADS1 mediates AA synthesis in CRC

The AA source in mammals is mainly dependent on self-synthesis. The endogenous synthesis of AA in CRC was investigated next. The endogenous synthesis of AA is controlled by several synthetic enzymes, including FADS1, FADS2, FADS3, ELOVL2 and ELOVL5, and is released by polylactic acid (PLA) enzymes. Initially, the expression of these enzymes in CRC were assessed using Gene Expression Omnibus (GEO) databasess. Only *FADS1*, but not other enzymes, was upregualted in CRC samples (Fig. 3a and Supplementary Fig. 3a-s). AA and eicosapentaenoic acid (EPA) are two products of FADS1 catalysis. EPA shows a tumor-suppressed role in many types of malignant tumors. The AA level was gradually upregulated in CRC process (Fig. 1a) and positively correlated with the *FADS1* expression (Fig. 3c). However, the EPA level showed no significant change and was not correlated with the *FADS1* expression in CRC (Fig. 3b, c). *FADS1* knockdown significantly decreased the AA level, as predicted, but no notable effect on the EPA levels in CRC cells was observed (Fig. 3d). Interestingly, the ratio of AA/EPA was significantly downregulated in sh*FADS1* CRC cells, compared with shNC CRC cells (Fig. 3e). Thus, upregulated FADS1 in CRC primarily affect the synthesis of AA rather than EPA. These results suggested that FADS1-mediated AA synthesis was a potentially important process in CRC.

### FADS1 exhibits CRC promotion by increasing gram-negative gut microbes

An in vivo orthotopic model was established using CRC cells and the organoid model (Fig. 4a and Supplementary Fig. 4a, b, 5a), and *FADS1* knockdown was found to significantly suppress orthotopic tumor growth (Fig. 4b, c and Supplementary Fig. 5b, c). However, in vitro and in subcutaneous tumors, *FADS1* knockdown had no obvious effect on cell proliferation (Supplementary Fig. 4c-e, 5e). This was quite similar to the AA pattern on CRC tumors. Therefore, we speculated that FADS1-AA axis may be a unified chain that promotes tumor growth by altering the intestinal microecology.

As expected, *FADS1* knockdown significantly decreased the proportion of gram-negative bacteria and increased gram-positive bacteria in the orthotopic model (Fig. 4d and Supplementary Fig. 5d). In detail, *FADS1* knockdown also significantly reduced *Es.coli*, *En.faecalis* and increased *Lactobacillus* (Fig. 4e and Supplementary Fig. 5d). Subsequently, we investigated whether gut microbes were indeed involved in tumor promotion of FADS1. Elimination of gut microbes almost completely abolished the FADS1 effect on tumor growth (Fig. 4b, c and Supplementary Fig. 5b, c). Next, we used aztreonam to eliminate gram-negative bacteria and vancomycin to eliminate gram-positive bacteria, and the aztreonam treatment showed a decreased tumor size, whereas the vancomycin treatment had no significant inhibitory role compared with non-antibiotic treatment in shNC tumor growth (Fig. 4f and Supplementary Fig. 4f, 5c). Further analysis revealed that shNC+vancomycin groups showed a similar gut microbiota composition with shNC groups, and shNC+aztreonam groups showed a similar gut microbiota composition with sh*FADS1* groups in the orthotopic tumor when the experiment was over (Supplementary Fig. 4g). In vitro and in subcutaneous tumors, Lipopolysaccharide (LPS) activation from *Es.coli* promoted cell proliferation in shNC CRC cells, but inhibited cell proliferation to some extent in sh*FADS1* CRC cells (Supplementary Fig. 6a-e). Adding AA exogenously rescued the decreased cell proliferation in sh*FADS1* CRC cells under LPS activation compared with shNC CRC cells (Supplementary Fig. 6a-e). Thus, the tumor-promoting effect of FADS1-AA axis was unified, and mediated by enriched gram-negative gut microbes in CRC.

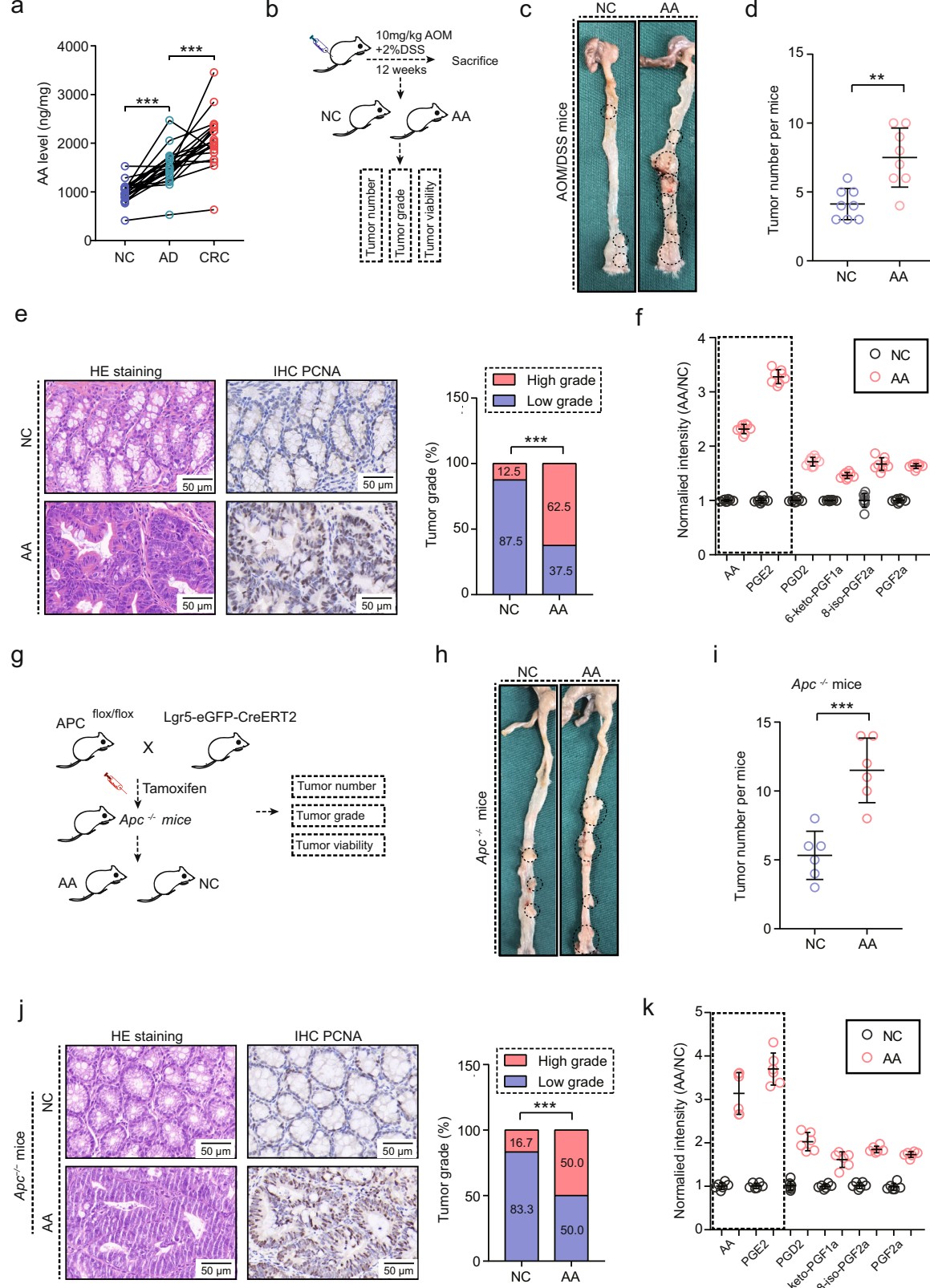

## FADS1 modulates the gram⁻ microecology via creating a high AA microenvironment

As mentioned above, FADS1-AA axis showed a unified pattern on gut microbes regulation. However, how AA involved FADS1 modulating the gram-negative microecology was still unknown. AA is stored in cell membranes of resting somatic cells in the form of phospholipids and is released into the cytoplasm and extracellular in response to various stimuli. In the orthotopic xenograft, the AA levels was significantly decreased in the interstitial fluid of tumor tissues in the sh*FADS1* group, compared with the shNC group (Supplementary Fig. 4h, 5f), indicating FADS1 played a role in creating a high AA microenvironment. Moreover, elimination of gut microbes with quadruple-

**Fig. 1 | The tumor-promoting effect of AA on CRC tumorigenesis. a** AA levels in paired NC, Ad and CRC samples, NC: normal intestinal epithelial tissues, Ad: colon adenoma, CRC: colorectal cancer. ***$p < 0.001$. ($n = 15$ per group, means ± s.d.; two-tailed paired $t$ test, $n = 3$ technical replicates). **b** CRC model established by 10 mg/kg of AOM induction (i.p injection) and 2% of DSS in drinking water using C57BL/6 J mice, NC: normal feeding, AA: arachidonic acid feeding ($n = 8$ mice per group). **c** Representative images of AOM/DSS-induced tumor in full-length colons ($n = 8$ mice per group). Black circles showed the tumor region. **d** Tumor number of AOM/DSS mice for NC and AA groups ($n = 8$ mice per group, means ± s.d.; two-tailed unpaired $t$ test, $n = 2$ biological replicates), **$p = 0.003$. **e** HE staining and PCNA expression in AOM/DSS tumors ($n = 8$ samples per group, 3 fields assessed per sample, Chi-square test). Scale bars, 50 μm. Tumor grade was independently assessed by two staff members of the pathology department, low grade and high grade represents low and high-grade intraepithelial neoplasia. ***$p < 0.001$. **f** AA and prostaglandins levels in tumor tissues in NC and AA groups in AOM/DSS-induced

mice ($n = 8$ samples per group, means ± s.d., $n = 3$ technical replicates). Liquid chromatography-mass spectrometry (LCMS) was used to detect the index levels. **g** Intestine-specific $Apc^{-/-}$ mice generated by crossing $Apc^{flox/flox}$ with $Lgr5$-$EGFP$-$IRES$-$CreERT2$ mice, NC: normal feeding, AA: arachidonic acid feeding ($n = 6$ mice per group). **h** Representative images of CRC tumors in intestine-specific $Apc^{-/-}$ mice ($n = 6$ mice per group). Black circles showed the tumor region. **i.** Tumor number of intestine-specific $Apc^{-/-}$ mice for NC and AA groups ($n = 6$ mice per group; means ± s.d., two-tailed unpaired $t$ test, n = 2 biological replicates). ***$p < 0.001$. **j** HE staining and PCNA expression in tumor tissues of intestine-specific $Apc^{-/-}$ mice ($n = 6$ samples per group, 3 fields assessed per sample, Chi-square test). Tumor grade was independently assessed. ***$p < 0.001$. Scale bars, 50 μm. **k** AA and prostaglandins levels in tumor tissues of intestine-specific $Apc^{-/-}$ mice. LCMS was used to detect the index levels. ($n = 6$ samples per group, means ± s.d.). Source data are provided in the Source Data file.

antibiotic administration, the AA showed a low levels and no statistical difference in the interstitial fluid of tumor tissues in the shFADS1 group and shNC group (Supplementary Fig. 4h, 5f), indicating gut microbes also involved the formation of high AA microenvironment. In vitro experiment, LPS (Es.coli) activation showed obviously increased AA level in the culture supernatant in a concentration-dependent manner in shNC cells, but only showed a slight elevation in shFADS1 cells (Fig. 3f). However, EPA levels remained constant regardless of FADS1 knockdown (Fig. 3g). The AA/EPA ratio was upregulated in the culture supernatant of shNC cells, but downregulated in the culture supernatant of shFADS1 cells with dependence on the concentration of LPS (Fig. 3h). These results suggested that high AA microenvironment and enrich gram-negative microecology was a mutually reinforcing process. Furthermore, we used aztreonam to eliminate gram-negative bacteria and vancomycin to eliminate gram-positive bacteria, and the aztreonam treatment showed a decreased AA level, whereas the vancomycin treatment had a roughly equivalent AA level compared with non-antibiotic treatment in the interstitial fluid of shNC tumor tissues in the orthotopic xenograft (Supplementary Fig. 4i, 5f). Moreover, as mentioned above, AA feeding contributed to an enriched gram-negative microecology (Fig. 2d-f). Together, FADS1-AA axis contributed to the creation of high AA microenvironment in CRC under intestinal flora activation, and in turn became the foundation for enriched gram-negative microecology.

### Increased gram-negative microbes contribute to PGE2 production
As mentioned above, LPS (Es.coli) activation showed drastically different results in shNC and shFADS1 CRC cells, indicating that LPS motivated a certain aspect of FADS1 function. Therefore, we further explored the deeper relationship between FADS1-AA axis and gut microbe alteration in CRC process. Initially, the gene set enrichment analysis (GSEA) was based on the *FADS1* expression in the TCGA dataset, including the mRNA sequence data from patients with CRC. Gene sets from the *FADS1* high-expression cases were enriched in AA metabolism (Fig. 5a). Prostaglandins, especially PGE2, are primary derivatives of AA metabolism. Consistent to AA feeding (Fig. 1f, k), after fecal transplant in AOM/DSS-induced and intestinal-specific $Apc^{-/-}$ mice, tumor tissues showed a higher PGE2 levels in stool-AA mice than in stool-NC mice (Fig. 5b). In the orthotopic model, shNC tumors showed notably higher PGE2 levels than shFADS1 tumors (Fig. 5c). Upon deletion of gut microbes, PGE2 levels were not statistically different in shNC and shFADS1 tumor (Fig. 5d). Furthermore, in vitro experiment, *FADS1* knockdown showed no obvious effect on the PGE2 levels in CRC cells (Fig. 5e). Upon LPS activation, shNC CRC cells showed significantly higher PGE2 levels than shFADS1 CRC cells (Fig. 5f). Adding exogenous PGE2 rescued the decreased cell proliferation in shFADS1 CRC cells, compared with shNC CRC cells, under LPS activation (Supplementary Fig. 6e, Supplementary Fig. 7a, b).

Prostaglandin-endoperoxide synthase 2 (PTGS2) and prostaglandin E synthase (PTGES) are key enzymes for PGE2 synthesis. The expressions of Ptgs2 and Ptges after fecal transplant were significantly upregulated in stool-AA mice compared with in stool-NC mice (Fig. 5g). The expression of PTGS2 and PTGES in orthotopic tumor tissues were notably higher in shNC than in shFADS1 mice, but they were not statistically different in the two groups with the deletion of gut microbes (Fig. 5h and Supplementary Fig. 7c,d). Exogenous LPS activation in vitro also upregulated the expression of PTGS2 and PTGES in CRC cells, however, *FADS1* knockdown did not directly affect the expression of PTGS2 and PTGES (Fig. 5i and Supplementary Fig.7e-h). Moreover, either the *PTGES* or the *PTGS2* knockdown inhibited LPS-induced cell proliferation, and adding PGE2 almost rescued this phenomenon (Fig. 6f and Supplementary Fig. 7i). Blocking PTGS2 by celecoxib also inhibited the promoting role of AA in tumor growth in AOM/DSS mice and intestine-specific $Apc^{-/-}$ mice (Supplementary Fig. 8d, e). Thus, FADS1-AA axis-induced enriched gram-negative microbes involves conversion of AA to PGE2, and promoted CRC cell proliferation.

### TLR4 involves in gram-negative microbes-mediated PGE2 production
Subsequently, we explored the molecular mechanism for gut microbe-mediated PGE2 production. Toll-like receptors (TLRs) are vital for sensing extracellular signals, specifically in the stimulation of gram-negative gut bacteria. The TLR4/MYD88 innate immune signaling pathway is primarily activated in response to gram-negative gut microbes. GSEA analysis, based on the *FADS1* expression using TCGA dataset, showed enrichment of TLR and inflammation response pathways in association with a high *FADS1* expression (Fig. 5a). PGE2 levels were significantly increased after LPS activation, and decreased by *TLR4* or *MYD88* knockdown in CRC cells (Fig. 6a, b and Supplementary Fig. 8a, b). FADS1 knockdown did not affect TLR4 and MYD88 expressions in vitro (Fig. 6c), indicating that FADS1 can not directly regulate the expressions of TLR4 and MYD88. Moreover, TLR4 or MYD88 knockdown significantly inhibited the elevated expression of PTGS2 and PTGES upon LPS activation (Fig. 6d, e). Functionally, *TLR4* or *MYD88* knockdown significantly inhibited LPS-induced cell proliferation in vitro and in vivo, and adding PGE2 rescued this phenomenon (Fig. 6f and Supplementary Fig. 8c). Blocking TLR4 with TAK-242 also inhibited the promoting role of AA in tumor growth in AOM/DSS mice and intestine-specific $Apc^{-/-}$ mice (Supplementary Fig. 8d, e).

### Upregulated FADS1 is an early event in CRC and predicts a poor prognosis
The role of FADS1 has been confirmed in vitro and in vivo, yet the relationship between FADS1 upregulation and clinical characteristics and prognosis of CRC patients remains unclear. First, we re-analyzed *FADS1* expression using GEO databases. The mRNA expression of *FADS1* exhibited an increasing trend from colon normal tissue to colon

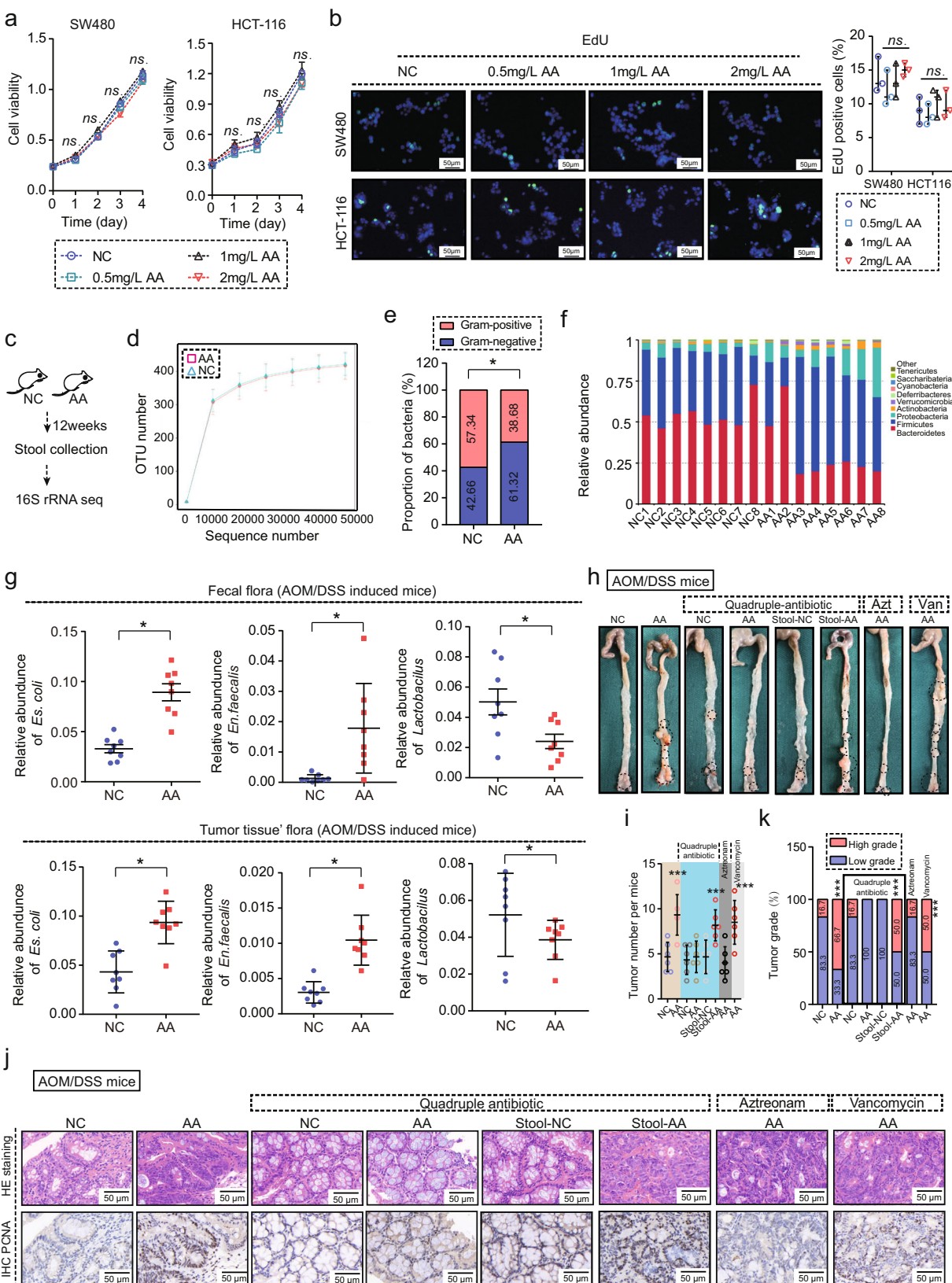

adenoma to CRC, but this was not true for other enzymes (Fig. 7a and Supplementary Fig. 9a-s). CRC patients with higher mRNA expression of *FADS1* showed shorter overall survival than patients with lower mRNA expression of *FADS1* (Fig. 7b). The protein expression of FADS1 also gradually increased during the progression of CRC from normal tissue to colon adenoma to CRC (Fig. 7c). Furthermore, the protein

expression of FADS1 was significantly upregulated and gradually increased from stage I to stage IV based on the tissue microarray from 392 CRC samples (Figs. 7d–f). The results demonstrated that FADS1 was upregulated in the early stage of CRC. CRC patients with higher protein expression of FADS1 also showed obviously shorter overall survival (Fig. 7g). An analysis of clinical information demonstrated that

**Fig. 2 | Gut microbes mediate tumor promotion via AA. a** The viability of SW480 and HCT-116 cells treated with different concentration (NC, 0.5 mg/L, 1 mg/L, 2 mg/L) of AA, and analyzed by CCK-8. ns indicates no statistical significance ($n = 5$ samples per group; means ± s.d., one-way repeated-measures ANOVA, $n = 3$ biological replicates). **b** EdU assay for SW480 and HCT-116 cells treated with different concentrations (NC, 0.5 mg/L, 1 mg/L, 2 mg/L) of AA ($n = 5$ samples per group, 3 fields assessed per sample, means ± s.d., two-tailed unpaired $t$ test). ns indicates no statistical significance. Scale bars, 50 μm. **c.** 16 s rRNA sequencing in stool of C57BL6J mice divided into NC and AA groups, NC: normal feeding, AA: arachidonic acid feeding ($n = 8$ mice per group). **d** Total OTU of gut microbes in the stool of C57BL6J mice, NC: normal feeding, AA: arachidonic acid feeding ($n = 8$ mice per group, means ± s.d., two-tailed unpaired $t$ test, $n = 3$ technical replicates). ns indicates no statistical significance. **e** Ratio of gram-negative bacteria to gram-positive bacteria in the stool of C57BL6J mice in NC and AA groups ($n = 8$ per group, *Chi* square test). *$p = 0.034$. **f** Relative abundance of bacterial OTUs in the stool of C57BL6J mice in NC and AA groups ($n = 8$ mice per group). **g** Relative abundance of *Es. coli*, *En. faecalis* and *Lactobacillus* in the stool and tumor tissue of AOM/DSS mice in NC and AA groups ($n = 8$ mice per group, means ± s.d., two-tailed unpaired $t$ test, n = 3 technical replicates). *$p = 0.016, 0.039, 0.018, 0.041, 0.012, 0.045$. **h** Representative images of AOM/DSS tumors with NC or AA feeding under different antibiotics treatment (Quadruple-antibiotics, 0.2 g/L of aztreonam, 0.1 g/L of vancomycin) ($n = 6$ mice per group). Black circles showed tumor region. **i** Tumor number of AOM/DSS-induced mice with NC or AA feeding under different antibiotics treatment ($n = 6$ mice per group; means ± s.d., two-tailed unpaired t test). ***$p < 0.001$(compared with NC group). **j** HE staining and PCNA expression in AOM-induced tumors with NC or AA feeding under different antibiotics treatment ($n = 6$ samples per group, 3 fields assessed per sample). Scale bars, 50 μm. Source data are provided in the Source Data file.

high protein expression of FADS1 was positively correlated with tumor size, T stage, and pathological stage (Fig. 7h). To determine the translational potential of FADS1, we used a FADS1 specific inhibitor, D5D-IN-326, to evaluate the anti-tumor effect in the orthotopic model, and the results showed that D5D-IN-326 treatment obviously inhibited the tumor growth and prolonged the overall survival of mice burdening with tumor (Fig. 7i, j).

Summarily, FADS1-AA axis in CRC cells that modulates the intestinal microecology of gram-negative microbes. Owing to these changes in the intestinal microecology, AA is converted to PGE2 via the TLR4/MYD88 pathway and eventually promotes CRC tumorigenesis (schematic diagram, Fig. 8).

## Discussion

AA shows both pro- and anti-tumorigenic activity[17]. On one hand, AA might be an apoptotic signal that regulates appoptotic processes[18]. On the other hand, AA also metabolizes to produce PGs, and many PGs, especially PGE2, are accumulated and promote inflammation and cancer progression[19–21]. We found that AA promoted the initiation and progression of CRC in AOM/DSS and intestine-specific $Apc^{-/-}$ mice. Conversely, exogenous AA had no significant pro-proliferation role in vitro and subcutaneous tumors. These inconsistent results indicated a special regulatory mechanism underlying tumor-promotion via AA in CRC.

Because of the presence of the gut microbial community, the tumor microenvironment of gastrointestinal cancer is unique[22]. This community is a complex mix of species, with the most abundance and diversity being accounted for by bacteria, and more than 90% of microbes exists in the colon and rectum[23]. Gut microbes and colorectum form a symbiotic relationship[24–26]. Gut microbes utilize the host's fatty acids and other substances, regulating the production of vitamins and short-chain fatty acids[26,27]. In turn, the host's fatty acids play a crucial regulatory role in the intestinal microecology[28,29]. Omega-3 polyunsaturated fatty acid (PUFA) supplementation statistically increases the abundance of the *Bifidobacterium* and *Oscillospira* genera, and reduces the abundance of *Coprococcus* and *Faecalibacterium*[30]. Giulia Piazzi reported that dietary eicosapentaenoic acid (EPA), an omega-3 PUFA, induced a significant increase in *Lactobacillus*, suppressing CRC tumor promotion and initiation[31]. In vivo, transplantation of the feces of patients with CRC into mice promoted AOM-induced colon tumor[32]. In our study, AA feeding increased the relative abundance of gram-negative bacteria in vivo and the modulated intestinal microbe mediated AA-induced CRC tumor development.

AA source in humans is dependent on self-synthesis, which is mainly mediated by fatty acid desaturases and fatty acid elongases. Fatty acid desaturase 1 (FADS1) and fatty acid desaturase 2 (FADS2) are rate-limiting enzymes for AA synthesis[33]. FADS1 catalyzes dimerization-γ-Linolenic acid (DHGLA) to produce AA and FADS2 involves in catalyzing linoleic acid to γ-linolenic acid[10]. Among the enzymes in the AA

metabolism, only the expression of FADS1 was significantly increased in our study. Similar to AA, FADS1 showed a tumor-promoting role in the orthotopic model, but not in the subcutaneous model or in vitro. However, it is still unclear whether the FADS1-AA axis is a uniform and effective pathway in CRC. AA and EPA are both catalytic products of FADS1, indicating that these two PUFAs compete in the reactions[34]. Unlike AA, EPA is a tumor-suppressed PUFA in many malignant tumors based on epidemiological studies[35]. Therefore, the PUFA type that dominates in synthesis may directly determine the impact of FADS1. In our study, *FADS1* knockdown reduced AA synthesis, but not EPA synthesis in CRC cells. AA is stored in cell membranes in the form of phospholipids in resting cells. Once stimulated (temperature, chemistry, physics and bacteria), AA is released by phospholipase A2 and becomes available for further metabolism[36]. The regulatory effect of FADS1 on gut microbiota needs to be elucidated to prove that FADS1-AA axis is an integral mechanism. Aberrant expression of host gene sharps the gut microbial composition. Synbindin ablation has been reported that enhanced the abundance of *Bacteroidetes*, *Actinobacteria* and decreased the abundance of *Firmicutes*, and *Dererribacteres*[37]. In our study, *FADS1* knockdown decreased the abundance of gram-negative bacteria in vivo, specifically reducing the numbers of *Escherichia coli* and *Enterobacter faecalis* and increasing *Lactobacillus*, consistent to the AA effect. Deletion of gram-negative bacteria also eliminated the tumor-promoting effect of FADS1. Therefore, FADS1-AA axis is a uniform and effective pathway in CRC by remodeling an intestinal microecology of enriched gram-negative bacteria.

Altered gut communities directly or indirectly regulate tumor growth and the progression of CRC by remodeling metabolism and immunity[38–41]. Other researchers have found similar phenomena that *FADS1* knockdown did not affect cell proliferation in CRC and breast cancer cells in general culture, however, with the addition of dihomo-γ-linolenic acid (DGLA, the precursor of AA), *FADS1* knockdown only showed an inhibited cell proliferation only in tumor cells with a high-expression cyclooxygenase-2 (PTGS2, COX2)[42,43]. These results further suggest that the tumor-promoting role of FADS1 depends on whether it could metabolize AA into downstream products. AA is an active substance in inflammatory signaling and is the direct precursor for prostaglandins[9]. Prostaglandins E2 (PGE2) produced by PTGS2 and PTGES has been reported to have a vital role in CRC tumorigensis and celecoxib, a selective inhibitor of PTGS2, reduced colon adenoma in $Apc^{Min/+}$ mice by blocking PGE2 synthesis[44,45]. In this study, we found that FADS1 had no direct regulatory role in PGE2 synthesis. However, FADS1-AA axis increases the intestinal microecology of gram-negative bacteria, and increased gram-negative bacteria induced the expressions of PTGS2 and PTGES, and promoted the PGE2 synthesis in CRC cells. Celecoxib treatment inhibited the tumor-promoting effect of FADS1 and AA.

Toll-like receptors (TLRs) are the primary signal transducers for human intestinal epithelial cells to sense activation by intestinal

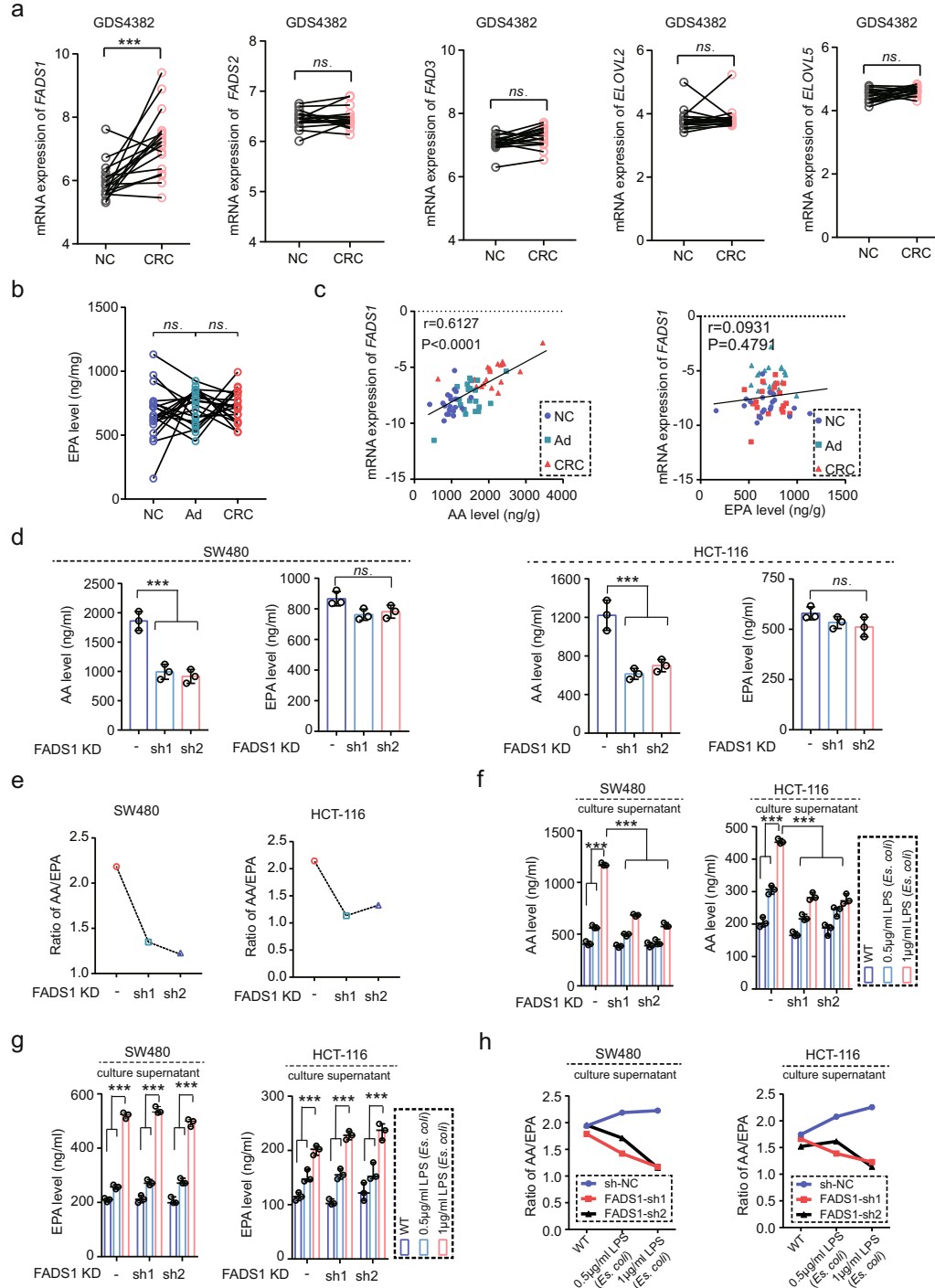

**Fig. 3 | FADS1 is upregulated in CRC and controls AA synthesis. a** The mRNA expression of *FADS1*, *FADS2*, *FADS3*, *ELOVL2*, and *ELOVL5* in 17 cases of paired NC and CRC samples from GDS4382, NC: normal colon epithelial tissue, CRC: color-ectal cancer. ***$p < 0.001$. ns indicates no statistical significance. **b** EPA levels in 15 cases of paired NC, Ad and CRC samples. ns indicates no statistical significance. ($n = 15$ samples per group, means ± s.d.; two-tailed paired *t* test, $n = 3$ technical replicates). **c.** Correlation between *FADS1* expression and AA or EPA levels in 15 cases of paired NC, Ad and CRC samples. The mRNA expression of *FADS1* was detected by q-PCR. r indicates correlation coefficient. *p* value indicates statistical difference ($n = 15$ per group, means ± s.d.; Pearson correlation test, n = 3 technical replicates). **d** Total AA and EPA levels in SW480 and HCT116 cells transfected with shNC or shFADS1. ***$p < 0.001$. ns indicates no statistical significance ($n = 3$ samples per group, means ± s.d.; two-tailed unpaired *t* test, $n = 3$ biological replicates). **e** AA/ EPA ratio in SW480 and HCT116 cells transfected with shNC or shFADS1 ($n = 3$

biological replicates). **f.** AA levels in culture medium from SW480 and HCT116 cells transfected with shNC or shFADS1, under activation with 0.5 µg/ml and 1 µg/ml of LPS, respectively ($n = 3$ samples per group, means ± s.d.; two-tailed unpaired *t* test, $n = 3$ biological replicates). CRC cells were cultured with FBS-free medium before LPS activation. ***$p < 0.001$. **g** EPA levels in culture medium of SW480 and HCT116 cells transfected with shNC or shFADS1, under activation with 0.5 µg/ml and 1 µg/ml of LPS, respectively ($n = 3$ samples per group, means ± s.d.; two-tailed unpaired *t* test, $n = 3$ biological replicates). CRC cells were cultured with FBS-free medium before LPS activation. ***$p < 0.001$. **h** AA/EPA ratio in culture medium of SW480 and HCT116 cells transfected with shNC or shFADS1, under activation with 0.5 µg/ml and 1 µg/ml of LPS, respectively ($n = 3$ biological replicates). CRC cells were cultured with FBS-free medium before LPS activation. Source data are provided in the Source Data file.

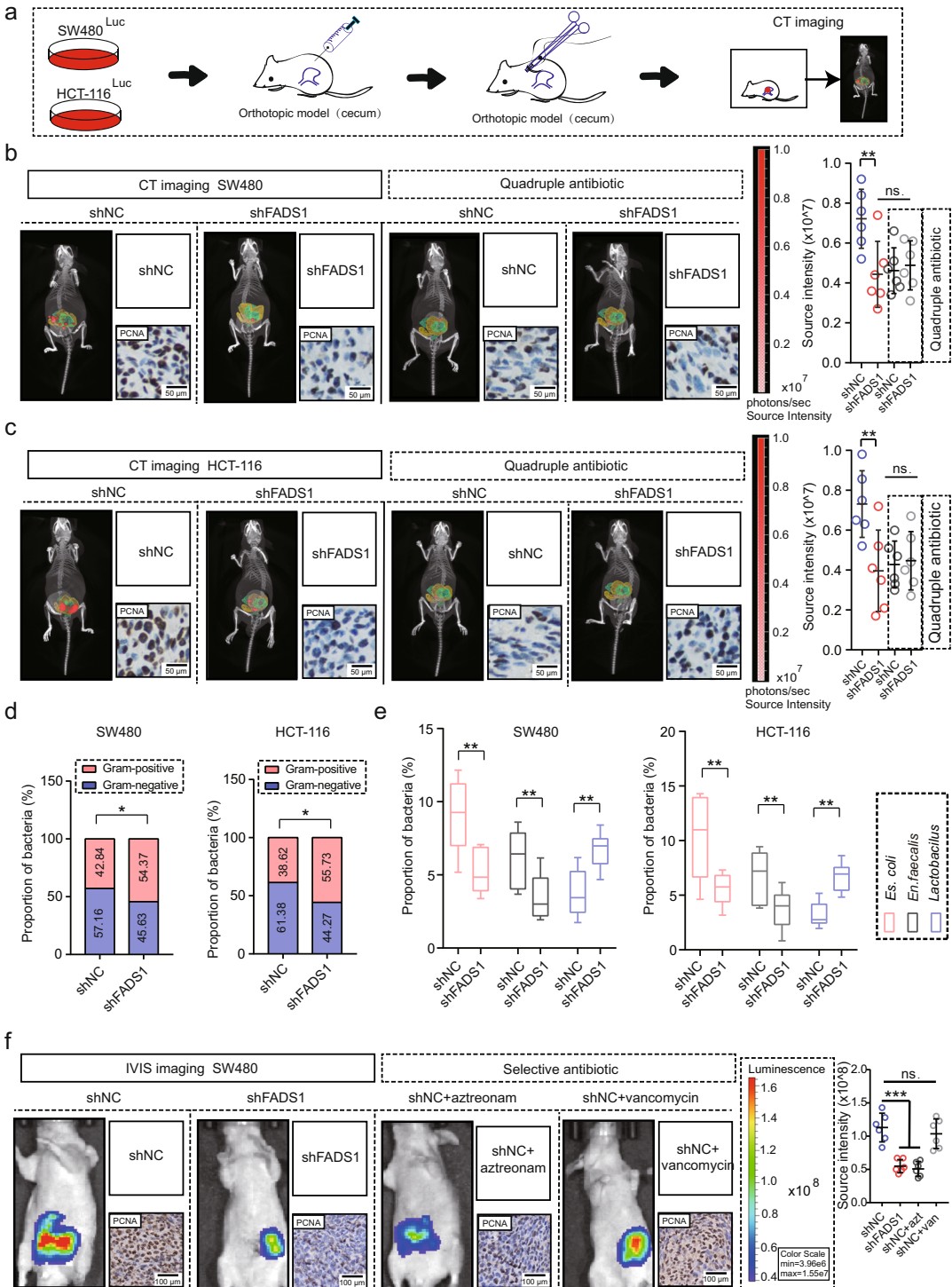

**Fig. 4 | Gut microbes mediate tumor promotion of FADS1 in CRC. a** An orthotopic tumor model of CRC was established by injecting SW480$^{Luc}$ cells or HCT116$^{Luc}$ cells into the wall of the cecum of BALB/C null mice. **b** Computed tomography (CT) with 3D organ reconstruction bioluminescence imaging of the orthotopic tumor of BALB/C null mice injected with shNC and sh*FADS1* SW480$^{Luc}$ cells (*n* = 6 mice per group, means ± s.d.; two-tailed unpaired t test). Gut microbes were deleted by quadruple antibiotics for two weeks. The yellow signal represents intestine, the green signal represents colon and the red signal represents tumor. Scale colour bar: $1×10^6$–$1.00×10^7$. **p = 0.007. **c** CT with 3D organ reconstruction bioluminescence imaging of orthotopic tumors in BALB/C null mice injected with shNC and *shFADS1* HCT116$^{Luc}$ cells (*n* = 6 mice per group, means ± s.d.; two-tailed unpaired t test). Gut microbes were deleted by quadruple antibiotics for two weeks. The yellow signal represents intestine, the green signal represents colon and the red signal

represents tumor. Scale colour bar: $1×10^6$–$1.00×10^7$. **p = 0.004. **d** Ratio of gram-negative and gram-positive bacteria in the orthotopic tumors of BALB/C null mice injected with shNC and sh*FADS1* cells (*n* = 6 mice per group, *Chi* square test, n = 3 technical replicates). *p = 0.037, 0.031. **e** Relative abundance of *Es. coli*, *En. faecalis* and *Lactobacillus* in the orthotopic tumor injected with shNC and sh*FADS1* cells (*n* = 6 mice per group, Box plots: Min to Max; two-tailed unpaired t test). **p = 0.008, 0.009, 0.005, 0.002, 0.004, 0.005. **f** IVIS imaging of the orthotopic tumor in the cecum of BALB/C null mice injected with shNC and sh*FADS1* SW480$^{Luc}$ cells (*n* = 6 mice per group, means ± s.d.; two-tailed unpaired t test), gut microbes were selectively deleted by 0.2 g/L aztreonam or 0.1 g/L vancomycin treatment for two weeks. Scale colour bar: $3.96×10^7$–$1.55×10^8$. ***p < 0.001, ns indicates no statistical significance. Source data are provided in the Source Data file.

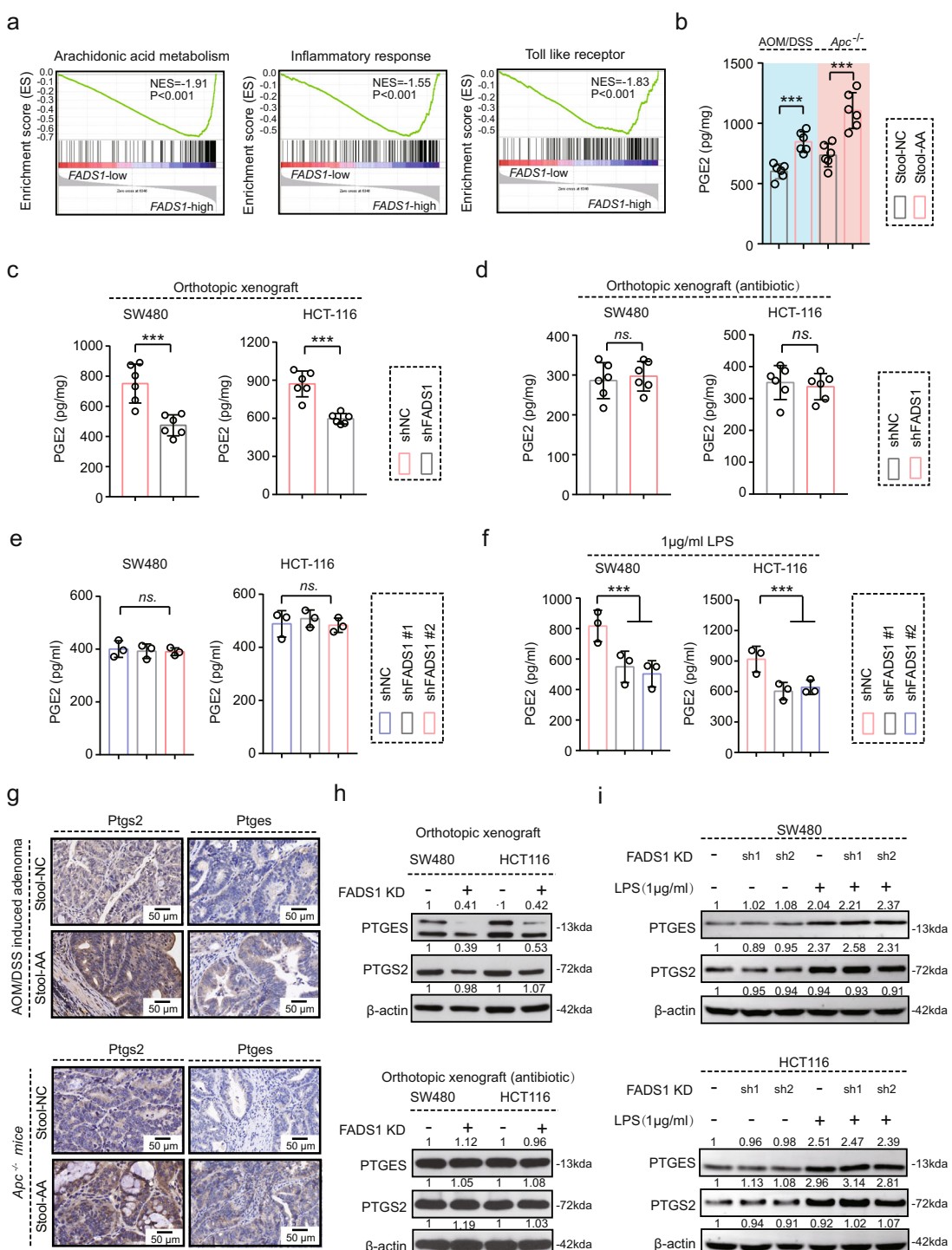

**Fig. 5 | Gram-negative gut microbes contribute to PGE2 production. a** GSEA analysis of *FADS1* expression in CRC, as evaluated from the TCGA dataset (440 CRC samples) and divided into high and low *FADS1* expression (220 samples each) groups. NES, normalized enrichment score, *p*, statistical significance. **b** PGE2 level in the tumor tissues of AOM/DSS or intestine-specific *Apc*$^{-/-}$ mice with fecal transplants of NC or AA feeding (*n* = 6 mice per group, means ± s.d.; two-tailed unpaired t test, *n* = 3 technical replicates). ***p* < 0.001. **c** PGE2 levels in the tumor tissues of the orthotopic tumors induced by SW480 and HCT116 cells transfected with sh*FADS1* or shNC (*n* = 6 mice per group, means ± s.d.; two-tailed unpaired t test, *n* = 3 technical replicates). ***p* < 0.001. **d** PGE2 level in the orthotopic tumors induced by SW480 and HCT116 cells transfected with sh*FADS1* or shNC, gut microbes eliminated with quadruple antibiotics (*n* = 6 mice per group, means ± s.d.; two-tailed unpaired t test, n = 3 technical replicates). ns indicates no statistical significance. **e** PGE2 levels in SW480 and HCT116 cells transfected with sh*FADS1* or shNC. ns

indicates no statistical significance. (*n* = 3 independent experiments). **f** PGE2 levels in SW480 and HCT116 cells transfected with sh*FADS1* or shNC, activated by 1 µg/ml of LPS (*n* = 3 per group, means ± s.d.; two-tailed unpaired t test, *n* = 3 technical replicates). ***p* < 0.001. **g** Representative IHC images of Ptges and Ptgs2 expression in tumor tissues of AOM/DSS or intestine-specific *Apc*$^{-/-}$ mice, with a fecal transplant from AA or NC feeding (*n* = 6 samples per group, 3 fields assessed per sample). Scale bars, 50 µm. **h** PTGES and PTGS2 expression in the tumor tissues of the orthotopic tumors transfected with sh*FADS1* or shNC, gut microbes eliminated with quadruple antibiotics (*n* = 6 mice per group). PTGES and PTGS2 expression were detected by western blot. The grayscale values of the WB bands are quantitatively represented (*n* = 3 technical replicates). **i** PTGES and PTGS2 expression in SW480 and HCT116 cells transfected with sh*FADS1* or shNC activated by 1 µg/ml of LPS. PTGES and PTGS2 expression were detected by western blot (*n* = 3 technical replicates). Source data are provided in the Source Data file.

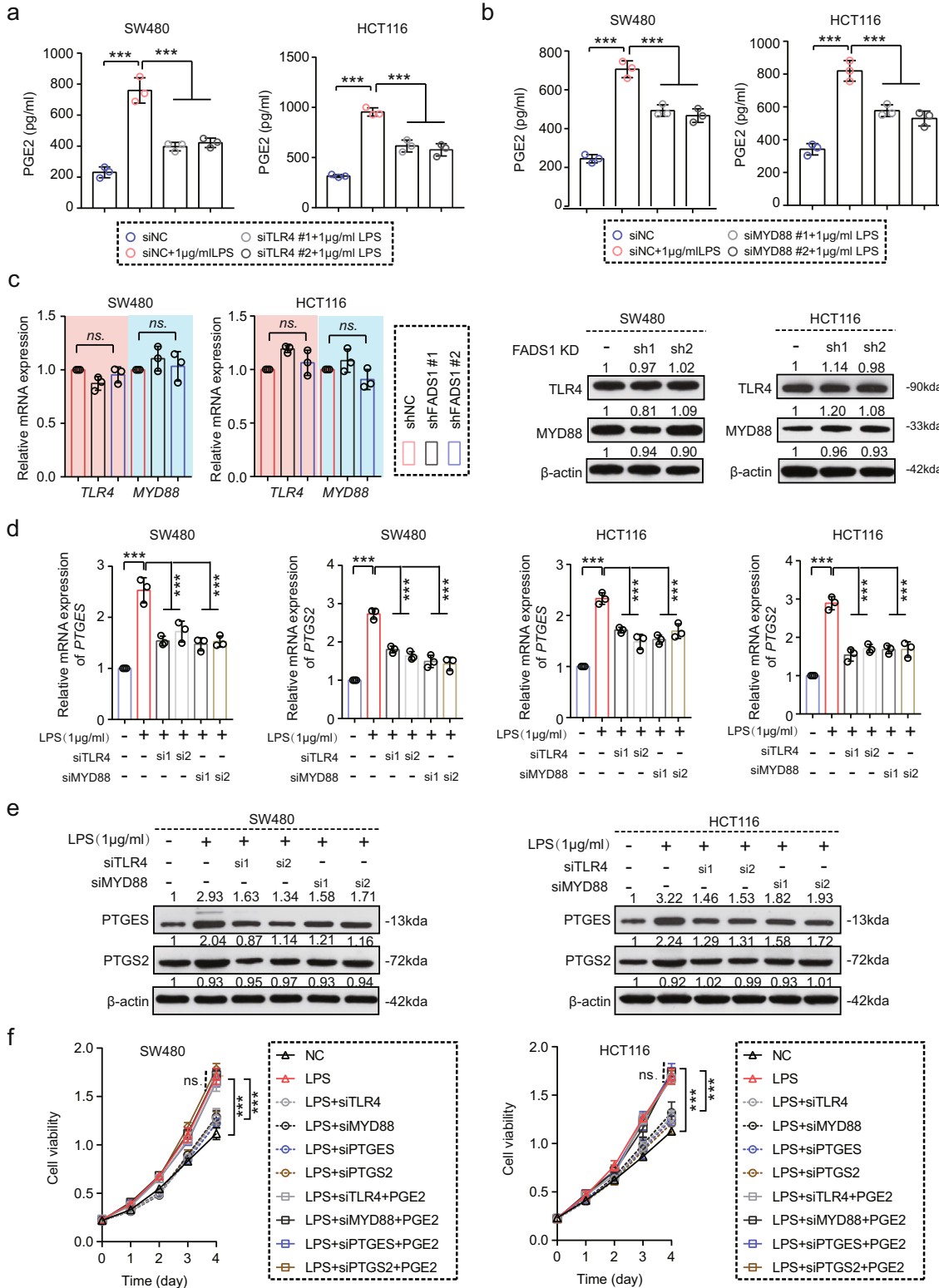

**Fig. 6 | The TLR4/MYD88 pathway is involved in the gut microbe-mediated PGE2 production. a** PGE2 levels in SW480 and HCT116 cells transfected with si*TLR4*, activated by 1 μg/ml of LPS ($n$ = 3 per group, means ± s.d.; two-tailed unpaired t test, $n$ = 3 biological replicates). ***$p$ < 0.001. **b** PGE2 level in SW480 and HCT116 cells transfected with si*MYD88*, activated by 1 μg/ml of LPS ($n$ = 3 per group, means ± s.d.; two-tailed unpaired t test, n = 3 biological replicates). ***$p$ < 0.001. **c** TLR4 and MYD88 expression in SW480 and HCT116 cells transfected with sh*FADS1* or shNC. TLR4 and MYD88 expression were detected by q-PCR and western blot. The grayscale values of the WB bands are quantitatively represented (n = 3 biological replicates). **d** The mRNA expression of *PTGES* and *PTGS2* in SW480

and HCT116 cells transfected with *siTLR4* and *siMYD88*, activated by 1 μg/ml of LPS ($n$ = 3 per group, means ± s.d.; two-tailed unpaired t test, $n$ = 3 biological replicates). ***$p$ < 0.001. **e** The protein expression of PTGES and PTGS2 in SW480 and HCT116 cells transfected with *siTLR4* and *siMYD8*8, activated by 1 μg/ml of LPS. The grayscale values of the WB bands are quantitatively represented ($n$ = 3 biological replicates). **f** Viability of SW480 and HCT116 cells transfected with *siTLR4*, *siMYD88*, *siPTGES*, *siPTGS2* and siNC, activated by 1 μg/ml of LPS, rescued by adding 1 mg/L of PGE2 ($n$ = 5 samples per group; means ± s.d., one-way repeated-measures ANOVA, $n$ = 3 biological replicates). ***$p$ < 0.001, ns indicates no statistical significance. Source data are provided in the Source Data file.

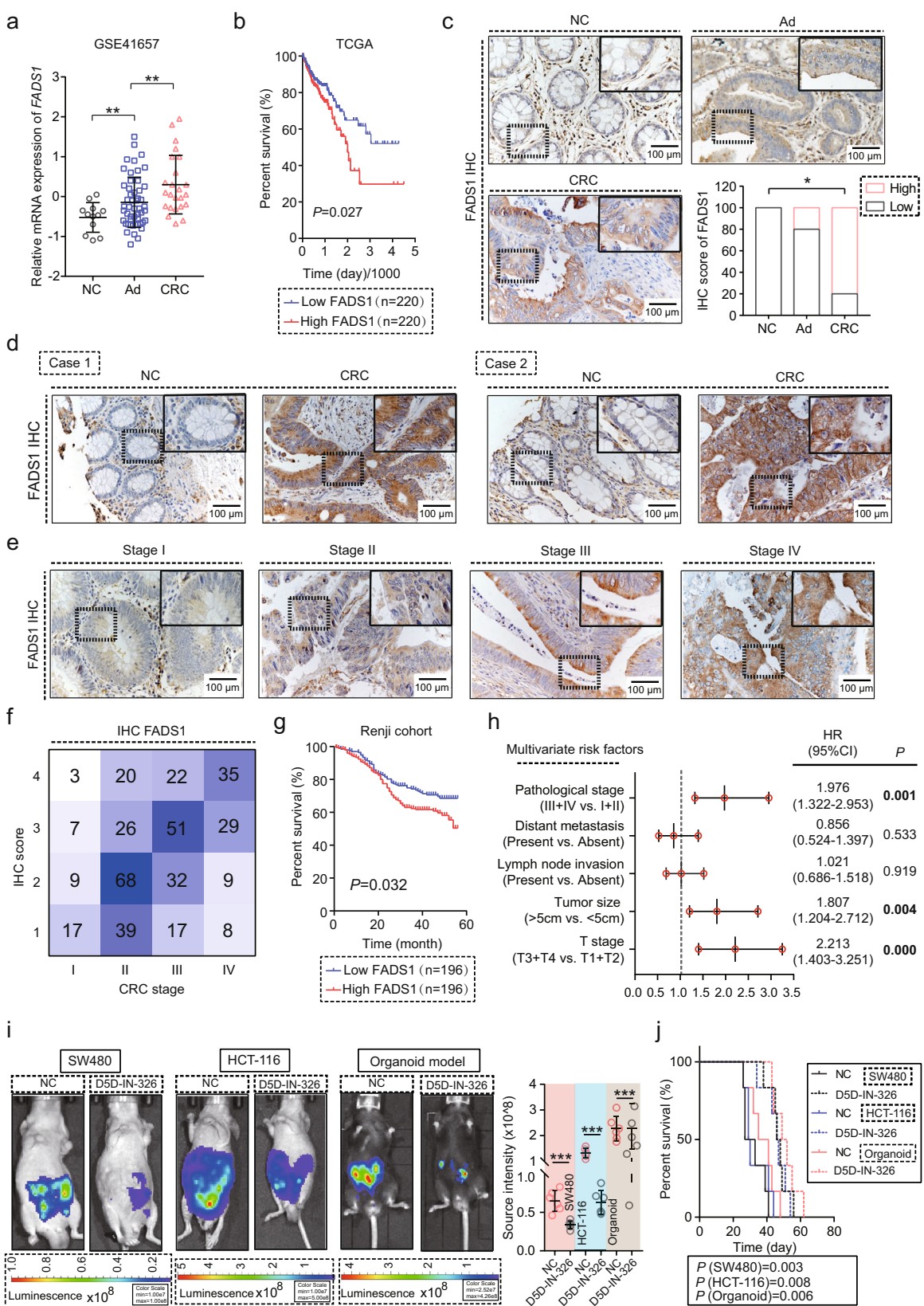

microorganism[46]. These receptors are prototypical pattern recognition receptors that recognize conserved microbial signature molecules, known as either pathogen-associated molecular patterns or microbe-associated molecular patterns (MAMPs)[47]. TLR4 is the primary receptor for gram-negative bacteria activation[39], and with the assistance of MYD88, extracellular signals further activate nuclear factor-κB (NF-κB) and mitogen-activated protein kinase cascades, leading to the transcription of inflammation-related target genes[48]. The TLR4/MYD88-NF-κB pathway is involved in PTGS2 gene transcription that promotes the conversion of AA to PGE2[49]. PTGES also

**Fig. 7 | FADS1 predicts a poor prognosis in CRC. a** The mRNA expression of *FADS1* in NC (*n* = 13 samples), Ad (*n* = 50 samples), and CRC (*n* = 25 samples) samples from GSE41657 (means ± s.d.; two-tailed unpaired t test), NC: normal colon epithelial tissue, Ad: colon adenoma, CRC: colorectal cancer. **p = 0.008, 0.007.
**b** Kaplan–Meier analysis of the relationship between the *FADS1* mRNA expression and prognosis of CRC patients from the TCGA dataset, *FADS1*-low: CRC patients with a low mRNA expression of *FADS1* (n = 220), *FADS1*-high: CRC patients with a high mRNA expression of *FADS1* (n = 220). *p* value showed the statistical difference.
**c.** Representative images of IHC staining of the FADS1 expression in matched NC, Ad, and CRC tissues (*n* = 6 samples per group, one-way repeated-measures ANOVA), NC: normal colon epithelial tissue, Ad: colon adenoma, CRC: colorectal cancer. IHC score was divided into Low and High groups, Low, score 1 or 2, High, score 3 or 4. *p = 0.029. **d** Representative images of IHC staining of the FADS1 expression in CRC and adjacent non-cancerous tissues of the tissue microarray (392 cases). Scale bars, 100 µm. **e** Representative images of IHC staining of the FADS1 expression in CRC tissues with different pathological stages. Scale bars, 100 µm.

**f** Correlation between the pathological stage and FADS1 expression in CRC and adjacent non-cancerous tissues of the tissue microarray (392 cases). Both variables are hierarchical data and rank sum test was used to statistical analysis. *p* = 0.034.
**g** Kaplan–Meier analysis of the relationship between the FADS1 protein expression and prognosis of CRC patients from the tissue microarray, FADS1-low: CRC patients with a low protein expression of FADS1 (n = 196), FADS1-high: CRC patients with a high protein expression of FADS1 (n = 196). *p* = 0.027. **h** Multivariate Cox regression analysis of clinicopathologic factors for the FADS1 expression applied in the Renji cohort (*n* = 392). HR, hazard ratio, 95% CI, 95% confidence interval. *p* = 0.001, 0.533, 0.919, 0.004, <0.001. **i** IVIS imaging of the orthotopic tumor treated with FADS1 inhibitor, D5D-IN-326 (2 mg/kg, p.o. for 2 weeks) (*n* = 6 mice per group, means ± s.d.; two-tailed unpaired t test). ***p < 0.001. Scale colour bar (SW480): $1 \times 10^7 - 1 \times 10^8$, Scale colour bar (HCT-116): $1 \times 10^7 - 5 \times 10^8$, Scale colour bar (organoid): $2.52 \times 10^7 - 4.26 \times 10^8$. **j** Survival analysis of the mice burden with the orthotopic tumors treated with D5D-IN-326 (*n* = 6 mice per group, Kaplan–Meier analysis). *p* = 0.003, 0.008, 0.006. Source data are provided in the Source Data file.

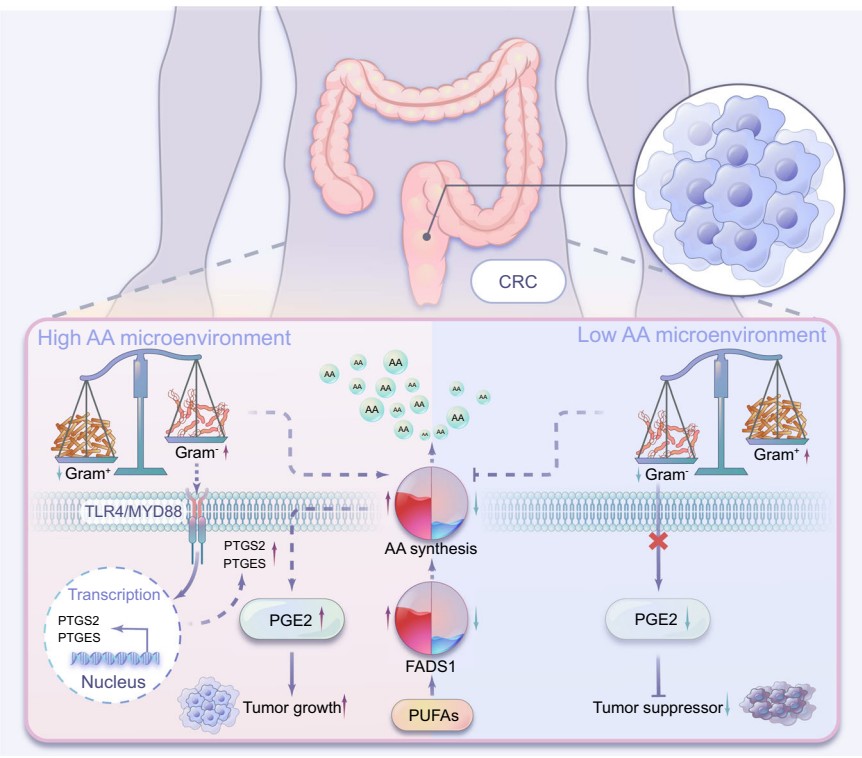

**Fig. 8 | The mechanism diagram of this study.** FADS1-AA axis in CRC cells that modulate the intestinal microecology of enriched gram-negative bacteria via creating a high AA microenvironment. Enriched gram-negative microecology accelerated AA converting to PGE2 and eventually promotes CRC tumorigenesis.

had transcriptional regulation via NF-κB[50]. Additionally, we found that TLR4/MYD88 pathway mediated the synthesis of PGE2 upon LPS activation.

Summarily, FADS1-AA axis in CRC cells modulate the intestinal microecology of enriched gram-negative bacteria via creating a high AA microenvironment, which promotes the conversion of AA to PGE2 and eventually encourages tumor growth. These complex interactions among tumor cells and intestinal microecology might be used to in characterize and develop potential therapeutic targets for the treatment of CRC.

## Methods
### Patients and samples
A total of 392 paraffin sections of CRC tissues and adjacent paired non-cancerous tissues were collected from the Department of Gastrointestinal Surgery, Renji Hospital, School of Medicine, Shanghai Jiao Tong University. Other information is shown in the supplementary materials.

### Animal experiments
All animal experiments were approved by the Research Ethics Committee of Renji Hospital and East China Normal University and adhered to local and national requirements for the care and use of laboratory animals. According to Ethics Committee, the maximal tumour size/burden permitted is 200mm³, which was strictly followed in this study. Modeling methods are shown in the supplementary materials.

### RNA isolation and real-time quantitative polymerase chain reaction (RT-qPCR)
RNA was extracted with Trizol (9109, TAKARA) and reverse transcribed using PrimeScriptTM (RR037A, TAKARA). The 18 S RNA was used as an internal control. The sequences of the primers used are shown in

Supplementary Table 1. Other information is shown in the supplementary materials.

## Immunohistochemistry (IHC)

IHC was performed as follows: all tissues were paraffin-embedded and cut into 4 μm thick sections. All sections were dewaxed with xylene and hydrated with alcohol. Sodium citrate was used for antigen retrieval, and 0.3% of hydrogen peroxide ($H_2O_2$) was used to block endogenous peroxidase. After blocking non-specific sites with bovine serum albumin, all the sections were incubated with an appropriate primary and secondary antibody. We used the 3,3-diaminobenzidine (DAB) kit (8801-4965-72, Thermo Fisher) for visualization, and hematoxylin was used to stain nuclei. All the sections were dehydrated with alcohol and sealed with neutral resin. The IHC staining score was calculated based on pixel intensity as follows: no staining, 1; weak staining, 2; moderate staining, 3; strong staining, 4. Pathological images were acquired using Leica Aperio ScanScope Console 12.3.

## Stool smear assay

Stool smear assays were used to measure the ratio of gram-negative to gram-positive bacteria. Stool was smeared on a glass slide and stained with gentian violet and fuchsine. Slides were examined with a 400× magnification and the percentage of gram-negative and gram-positive bacteria was calculated.

## Metabolite analysis

AA (NBP2-59872, Novus Biologicals), eicosapentaenoic acid (LM-11519-ES, Lianmai Biological Technology Co., Ltd.), and prostaglandin E2 (KGE004B, Novus Biologicals) were assessed using detection kits following the kit's instructions. Other information is shown in the supplementary materials.

## Lentivirus and siRNA transfection

Short hairpin RNAs (shRNAs) were transfected into CRC cell or organoid cell via lentivirus to generate shFADS1, shFads1, shTLR4 and shMYD88; shNC was used as the negative control. siRNAs for TLR4, MYD88, PTGES, and PTGS2 were purchased from GenePharma (Shanghai GenePharma Co., Ltd., Shanghai, China). All sequences are shown in Supplementary Table 1. Other information is shown in the supplementary materials.

## 16 S rRNA sequencing

16 S rRNA is a component of the 30 S small subunit of the prokaryotic ribosome. The conserved region is shared by bacteria, and the highly mutated region is race-specific. 16 S rDNA primers were designed and amplified and high-throughput technology was used. The types of microorganisms were thus identified. The experimental process was conducted by HiSeq 2500, and divided into DNA quality testing, 16 S rDNA library preparation, library testing, sequencing, and data analysis. All data were uploaded to National Center for Biotechnology Information (PRJNA762520).

## Statistics and reproducibility

**Statistical analysis.** Data are presented as means ± standard deviation (SD). Statistical analyses used SPSS 20.0 (Chicago, IL, USA) and GraphPad Prism 7 software. The threshold for statistical significance was assumed to be $p < 0.05$.

Additional experimental methods and information are provided in the supplementary materials file.

## Reporting summary

Further information on research design is available in the Nature Portfolio Reporting Summary linked to this article.

## Data availability

The 16 S rRNA sequence data generated in this study were deposited in the NCBI Sequence Read Archive (SRA) database under the accession code PRJNA762520. Publicly available data used in this work can be acquired from the TCGA Research Network portal and Gene Expression Omnibus (GDS4382 and GSE41657, (https://www.ncbi.nl m.nih.gov/geo/query/acc.cgi?acc=GDS4382; https://www.ncbi.nlm.nih. gov/geo/query/acc.cgi?acc=GSE41657). All data generated in this study are provided with this paper as Source data file. Source data are provided with this paper.

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

## Acknowledgements

This study was supported by the National Natural Science Foundation of China (82072671), Research Fund for medical engineering cross project of Shanghai Jiao Tong University (No. ZH2018ZDA08) hosted by Qing Xu; National Natural Science Foundation of China (92168111, 82230087), the Shanghai Municipal Education Commission—Gaofeng Clinical Medicine Grant Support (20181708), Program of Shanghai Academic/Technology Research Leader (19XD1403400), Shanghai International Science and Technology Cooperation Fund (18410721000), Excellent Academic Leader of Shanghai Municipal Health Bureau (2018BR32), Medicine and Engineering Interdisciplinary Research Fund of Shanghai Jiao Tong University (YG2021ZD08), Innovative research team of high-level local universities in Shanghai (SHSMU-ZDCX20210802), Shanghai Pilot Program for Basic Research - Shanghai Jiao Tong University (21TQ1400225) hosted by Zhigang Zhang; China Postdoctoral Science Foundation (2018M640403), National Natural Science Foundation of China (81701945) hosted by Xu Wang; National Natural Science Foundation of China (82073023, 81871923), Shanghai Municipal Education Commission—Gaofeng Clinical Medicine Grant Support (20191809) hosted by Jun Li; National Natural Science Foundation of China (82103357), Shanghai Sailing Program (21YF1445200), Natural Science Foundation of Shanghai (21ZR1461300) by Lipeng Hu.

## Author contributions

X.C.J., G.L., J.C.H. and H.L.P. carried out the molecular genetic studies, participated in the sequence alignment and in vitro and in vivo experiment, and these 4 authors contributed equally. X.Q., Z.Z.G., Z.X.L., X.H.B. and L.J. participated in the design of the study and supervised the work. L.Q., S.L.C., L.Y. and Z.H. conceived of the study, and participated in its design and coordination and helped to draft the manuscript. All authors read and approved the final manuscript.

## Competing interests

The authors declare no potential conflicts of interest.
