## [Peer Review File · Nature Communications]

FADS1-arachidonic acid axis enhances arachidonic acid metabolism by altering intestinal microecology in colorectal cancerREVIEWER COMMENTS

Reviewer #1 (Remarks to the Author):

In this manuscript by Xu et al., the authors investigate the interplay between the FADS1/arachidonic acid (AA) axis and the gut microbiome in colorectal cancer (CRC). Whilst some interesting observations are presented with regards to the role of AA in regulating intestinal microecology, most of the work that is presented is correlative and the interpretation of the acquired data suffers from almost being one-directional. There are several enzymes in AA metabolism that could play a role in some of the observed phenotypes and the efficacy of FADS1 targeting is quite low. Furthermore, most of the data presented on the role of TLR4/MYD88-dependent, LPS-induced synthesis of PGE2 simply validate already published data. Finally, this manuscript would benefit from more detailed explanation of the results, and some editing help from someone with proficiency in English and scientific writing. The authors should address the following specific comments:

Major comments:

Figure 1: The authors do not provide any information on what concentration of AA was provided in the diet, nor what is the control diet or whether it is isocaloric? Given that linoleic acid (LA) is the major dietary polyunsaturated fatty acid in the Western diet and is a metabolic precursor to AA, it would be interesting to demonstrate whether LA feeding can also significantly promote CRC development. The authors should also measure the levels of AA and downstream prostaglandins in the adenomas formed with or without dietary AA or LA supplementation using LC-MS analysis.

Figure 2: Fig.2A, In contrast to previous studies in CRC (Habbal et al., 2009, World J Gastroenterol) or breast (Koundouros et al., 2020, Cell), the authors demonstrate AA treatment having no obvious effects on the cell proliferation of SW480 or HCT116 cells. However, given that these experiments are performed in regular culture condition, it is unclear what is the concentration of AA, or other fatty acids where AA can be obtained from, in the FBS that the cells are cultured in. These could already be at saturated levels even before the supplementation of AA. To really assess the effects of exogenous sources of AA in cell proliferation, the authors should repeat this experiment in fatty acid free conditions with or without the supplementation of AA.

Fig.2D-G and Fig.S1A-S1C, The effect of AA at increasing the proportion of gram-negative bacteria and decreasing gram-positive bacteria is intriguing, but the exact mechanism is unclear. To further shed some light in this, the authors should measure the microbiome in the faeces of Apc WT mice following the addition of AA. Furthermore, the authors should establish what is the effect of antibiotic administration in CRC progression in regular diet fed Apc^{-/-} mice without the supplementation of AA. An experiment combining fecal transplantation and antibiotic treatment in regular diet fed mice would also provide further confidence on the proposed model.

Figure 3: There are several enzymes contributing to AA metabolism including the PLA enzymes that regulate its release from phospholipids, yet the gene expression analysis presented in Fig.3A is only performed in the genes that are involved in AA synthesis. The gene expression analysis in GDS4382 should be extended to all the genes that play a role in AA metabolism and the results should be validated in the GSE41657 dataset presented in Fig 7.

Figure 4: It is intriguing that the tumour suppressive effects of FADS1 appear to be tissue specific. Suppression of FADS1 induces ROS generation, and it would be useful to check if the effects of FADS1 knockdown in the microbiome are driven by an increase in redox stress, and if so, whether they can be rescued by treatment with antioxidants. To further establish the cell autonomous vs non-cell autonomous role of AA metabolism in CRC growth it would be useful to assess if AA affects tumour growth of subcutaneous tumours and whether LPS treatment on its own increases cell proliferation in vitro and subcutaneous tumour growth in vivo.

Fig.4F-G: To highlight the greater impact of microbiome removal in tumour development in FADS1 KD cells, it would be useful to plot the changes in tumour growth that can be attributed to the treatment with antibiotics only. Comparing CT imaging source intensity or PCNA staining between

experiments 4B-C and 4F-G would provide more confidence that the effect is not only due to the antibiotic treatment. It would also be useful to report if any effect on mouse weight was observed.

Fig.S3A-B: FADS1 KD appears to decrease cell proliferation upon LPS activation. However, by comparison with tumour weight and volume in Fig.S2D-E, it seems that LPS leads to an increase in tumour volume and weight in shNC condition rather than causing a decrease in FADS1 KD (providing that the duration of experiments is similar, which is not mentioned). It would be useful to repeat the experiments in S3A and S3B in fatty acid free conditions and assess if addition of AA can rescue the decrease in cell proliferation that is observed in FADS1 KD cells in the presence of LPS treatment.

Figure 5: The authors should suggest at least an explanation as to why FADS1 KD in vitro results in a decrease in AA but no changes in PGE2, and what is the proposed mechanism of decreased PTGS and PTGES expression following FADS1 KD.

Figure 6: Fig.6F To attribute the increase in cell proliferation solely to the increase of AA-mediated production of PGE2, it would be interesting to see if addition of PGE2 can rescue TLR4/MYD88 KD-mediated inhibition of cell viability after LPS treatment in SW480 and HCT116 in vitro. The authors should also check if inhibition of the TLR4/MYD88 signalling pathway in vivo rescues the effect of AA addition at increasing tumour growth.

Minor comments:

Figure 1A: It is not clear how was Arachidonic Acid (AA) levels measured? The authors should provide more information in the materials and method section on how AA was extracted from samples, whether these are frozen or paraffine embedded etc.

Figure 1F: mentions tumour grade but this was not assessed for the APC -/- model following AA or NC diet.

Figure 2F: The histogram represents distribution for NC=8 and AA=8, however the Figure legend for 2c mentions n=6 for both groups. If this is from the same experiment, the number of mice per group need to be revised.

Figure 2J and supplementary figure 2F are exactly the same figure (Tumor number per mice). Was the number of tumours identical in both experiments with the two different models or is this a mistake when compiling the figures?

Figure 3B: legend in text line 252 mentions AA but the figure is about EPA. AA increase was in figure 1A. Text or figure legend should be corrected.

Line 286: observations are made for figure 4F and G but only 4G is mentioned in the text.

Line 719: microbeq2-mediated in figure legend.

Line 430: the authors wrote that FADS1 KD induces significantly reduced proliferation of CRC cells in vivo and in vitro. They then write in line 434 that FADS1 KD does not show significant inhibition of cell proliferation. This should be corrected.

Reviewer #2 (Remarks to the Author):

In the manuscript "The FADS1/AA axis enhances AA metabolism by altering intestinal microecology in CRC" Xu et al. report a finding that AA increase/feeding positively influences CRC growth. THE manuscript is potentially interesting, however, the mechanistic data is rather weak and mostly based on in vitro cell line experiments. The authors overinterpret some of the findings based on inadequate in vivo models.

- the data on the shift to gram neg bacteria following AA feeding is interesting. The authors fail to proof that in in vivo experiments. The use of antibiotic treatment generally abolishes CRC growth - therefore proofing the point the reviewers make that the AA effects is mediated by gram neg bacteria in this set up is not possible. This set up is used in Fig 2 and 4 and not valid to answer this effectively

- for in vivo experiments the authors use immunocompromised mice and human cells lines. This set up does not justify the claim of the authors. Organoid models using immunocompetent mice

need to be used.

- showing that prostaglandin E2 levels depend on LPS activation of TLR4 is a nice observation in vitro. However, this is not proven in vivo and therefore the data is overinterpreted.

Reviewer #3 (Remarks to the Author):

In this manuscript, the role of the FADS1/AA axis in colorectal cancer (CRC) via modulating gut microbiota was explored. FADS1 knockdown decreased the ratio of Gram-negative and Gram-positive bacteria which impacted AA metabolism and decreased the production of a primary AA derivative prostaglandin E2 (PGE2) through downregulation of PTGS2 and PTGES. However, PGE2 is known, through numerous studies in humans and mice, to promote colon carcinogenesis. This is the basis for the anti-colon cancer effects of NSAIDs like sulindac. The gut microbiota also influences colon cancer in several experimental models. Thus, the novelty of this study is lacking. The authors need to determine the mechanism by which the gut microbiota changes and how this change affects PGE2.

Comments:

1. FADS1 and FADS2 genes were found highly expressed in colon tumor tissue compared with normal colon tissue in a previous genome-wide association study of CRC in East Asians (PMID: 24836286). However, FADS2 expression showed no change in the present study. Explain the discrepancy?
2. AA was reported to regulate apoptosis. The TUNEL assay should be performed to measure apoptosis in cells and the mouse models.
3. Since LPS was widely used in cell experiments, the colitis-related CRC murine model (AOM-DSS) should be applied instead of the AOM model.
4. Images of full-length colons need to be shown instead of trimmed colons.
5. Under LPS treatment, FADS1 knockdown decreased PGE2 production in vitro (Figure 5F). However, expression of PTGES and PTGS2 were not significantly affected at both the mRNA and protein levels in vitro (Figure 5I, S4C-F). Why did PGE2 production decrease while expression of the core synthetic enzymes PTGES and PTGS2 were not changed?
6. Line 212: "...a spontaneous model of CRC by hybridizing Apc..." should be "a spontaneous model of CRC by crossing Apc".
7. Line 236: "Gut microbial elimination by antibiotic administration completely inhibited the tumor-promoting activity of AA (Figure 2H, Figure S1D)". Tumor numbers should be shown and statistically analyzed.
8. Line 250: "AA and eicosapentaenoic acid (EPA) are two products of FADS1 catalysis. AA level was gradually upregulated in CRC and positively correlated with FADS1 expression (Figure 3B)." This should be Figure 3C. "However, EPA level showed no significant change and was not correlated with FADS1 expression (Figure 3C)." This should be Figure 3B&C.
9. Line 261, Figure 3F: "We found that lipopolysaccharide (LPS) activation from Escherichia coli significantly increased AA levels in culture supernatant in a concentration-dependent fashion in shNC cells, but not in shFADS1 cells (Figure 3F)." In shFADS1 cells, LPS still increased AA levels, just to a less extent compared to shNC cells. Besides, instead of labeling as "- + +" on x-axis, it should be labeled as "- sh1 sh2", since two shRNAs were used for each gene. Moreover, the knockdown efficiency of shFADS1, shTLR4, shMDY88, shPTGES, shPTGS2 needs to be determined by qPCR and WB.
10. Line 281: The mechanism by which knockdown of FADS1 reduced Escherichia coli and Enterococcus faecalis, and increased Lactobacillus needs to be determined.

11. Line 316, Figure 5I: "FADS1 knockdown did not affect PTGS2 and PTGES expression in vitro". It looks like PTGES was decreased by FADS1 KD in SW480 cells. All WB bands need to be quantified in this manuscript.
12. To prove that FADS1/AA axis affect CRC development via the PTGS2/PTGES-PGE2 pathway, rescue experiments should be performed. For example, 1. overexpress PTGS2/PTGES in shFADS1 cells and measure xenograft growth, 2 supplement PGE2 in shFADS1 cells or mouse models. 3. delete PTGS2/PTGES /inhibit PGE2 pathway in AA treated mice.
13. Line 333: "Prostaglandin E2 levels significantly decreased after LPS activation (Figure 6A, 6B)". Should be "Prostaglandin E2 levels were significantly increased after LPS activation, and decreased by TLR4 or MYD88 KD".
14. What are the mechanisms by which the TLR4/MYD88 pathway regulates t expression of PTGS2 and PTGES? Is the TLR4/MYD88 pathway activated under CRC, or with AA treatment?
15. Transplantation of Escherichia coli, Enterococcus faecalis or Lactobacillus in mice should be carried out in order to determine the role of gut microbiota in CRC.
16. To determine the translational potential of this study, a FADS1 inhibitor such as 2-(2,2,3,3,3-pentafluoropropoxy)-3-[4-(2,2,2-trifluoroethoxy) phenyl]-5,7-dihydro-3H-pyrrolo[2,3-d]pyrimidine-4,6-dione (PMID: 31488602), should be applied to xenograft or PDX mice.
17. Supplementary materials and methods: "Five million SW480 or HCT116 cells were injected into the left armpit of each null mouse." Mice do not have armpits.
18. A schematic diagram should be shown that summarizes the results.
19. The manuscript needs to be thoroughly proofread, as there are many typos and grammatical mistakes.

Point-By-Point responses:

Reviewer #1 comments:

In this manuscript by Xu et al., the authors investigate the interplay between the FADS1/arachidonic acid (AA) axis and the gut microbiome in colorectal cancer (CRC). Whilst some interesting observations are presented with regards to the role of AA in regulating intestinal microecology, most of the work that is presented is correlative and the interpretation of the acquired data suffers from almost being one-directional. There are several enzymes in AA metabolism that could play a role in some of the observed phenotypes and the efficacy of FADS1 targeting is quite low. Furthermore, most of the data presented on the role of TLR4/MYD88-dependent, LPS-induced synthesis of PGE2 simply validate already published data.

Finally, this manuscript would benefit from more detailed explanation of the results, and some editing help from someone with proficiency in English and scientific writing.

The authors should address the following specific comments:

Major comments:

Figure 1: The authors do not provide any information on what concentration of AA was provided in the diet, nor what is the control diet or whether it is isocaloric? Given that linoleic acid (LA) is the major dietary polyunsaturated fatty acid in the Western diet and is a metabolic precursor to AA, it would be interesting to demonstrate whether LA feeding can also significantly promote CRC development. The authors should also measure the levels of AA and downstream prostaglandins in the adenomas formed with or without dietary AA or LA supplementation using LC-MS analysis

Response:

Thank you very much for your careful review of our manuscript. In our study, we used AIN-93G diet for the control diet and diet enriched with AA-FFA 1.5% for AA feeding,

and 2 diets were isocaloric. Detailed information was shown in Supplementary materials (Table S2). For demonstrating whether LA feeding can also significantly promote CRC development, we used AIN-93G diet for the control diet and diet enriched with LA-FFA 3.5% for LA feeding (response to reviewer Table 1). 2 diets were isocaloric. LA feeding can also significantly promote CRC development in AOM/DSS mice (response to reviewer Figure 1a, b). And with dietary AA or LA supplementation, the AA and prostaglandin E2 levels showed the most significant elevation than other prostaglandins in colon tumors measured by LC-MS analysis as shown in Fig 1f, k and response to reviewer Figure 1c.

Table S2 AA feed formula for mice

	Control	1.5% AA
Casein	205	205
Corn starch	368	368
Dextrin	135	135
Sucrose	102	102
Soybean oil	72	72
Cellulose	51	51
Minerals	36	36
Vitamin	10	10
L-cystine	3	3
Choline tartrate	3	3
TBHQ	0.01	0.01
Cocoa butter	15	0
Arachidonic acid	0	15
Total(g)	1000	1000

response to reviewer Table 1 LA feed formula for mice

	Control	3.5% LA
Casein	210	210
Corn starch	330	330
Dextrin	139	139
Sucrose	105	105
Soybean oil	74	74
Cellulose	53	53
Minerals	37	37

Vitamin	11	11
L-cystine	3	3
Choline tartrate	3	3
TBHQ	0.01	0.01
Cocoa butter	35	0
Linoleic acid	0	35
Total(g)	1000	1000

response to reviewer Figure 1: a. Colon adenoma model established by 10 mg/kg AOM induction (i.p injection) and 2% DSS in drinking water using C57BL/6J mice, NC: normal feeding, n=6, LA:

linoleic acid feeding, n=6. b. Representative images of HE staining and PCNA expression in AOM/DSS-induced adenoma. c. LA and PGs levels in colon tumor tissues measured by LC-MS analysis.

Fig 1f, k: f. AA and prostaglandins levels in tumor tissues in NC and AA groups (n=8) in AOM/DSS-induced mice measured by LC-MS analysis. k. AA and prostaglandins levels in tumor tissues in NC and AA groups (n=6) in intestine-specific *Apc*^{-/-} mice measured by LC-MS analysis.

Figure 2: Fig.2A, In contrast to previous studies in CRC (Habel et al., 2009, World J Gastroenterol) or breast (Koundouros et al., 2020, Cell), the authors demonstrate AA treatment having no obvious effects on the cell proliferation of SW480 or HCT116 cells. However, given that these experiments are performed in regular culture condition, it is unclear what is the concentration of AA, or other fatty acids where AA can be obtained from, in the FBS that the cells are cultured in. These could already be at saturated levels even before the supplementation of AA. To really assess the effects of exogenous sources of AA in cell proliferation, the authors should repeat this experiment in fatty acid free conditions with or without the supplementation of AA

Response:

Thanks for the reviewer's reminding. Endogenous synthesis of AA is controlled by

several synthetic enzymes, including FADS1, FADS2, FADS3, ELOVL2 and ELOVL5, and is catalyzed and released by polylactic acid (PLA) enzymes. In the previous study (Koundouros et al., 2020, Cell)⁽¹⁾, it has reported that AA was increased in breast cancer cells by cPLA2. This “AA” may be the free AA. The free AA could be metabolized to downstream products, like PGE2, which play an important role in carcinogenesis. This is quite consistent with our point of view. In our study, FADS1 upregulation promoted the synthesis of AA. By regulating the intestinal microbial environment, it promotes AA release and the metabolism of AA into PGE2, which promotes the progress of CRC. As suggested by the reviewer, we repeated CCK-8 and EdU assay with different AA concentrations in fatty acid free medium (Detailed information was showed in Supplementary materials: cell culture). The results also showed that AA treatment had no obvious effects on the cell proliferation of SW480 and HCT116 cells (Supplementary Fig.1a-c).

Supplementary Fig 1a-c: a. The viability of SW480 and HCT116 cells treated with different concentration (NC, 0.5mg/L, 1mg/L, 2mg/L) of AA, cultured with fatty acid free FBS, and analyzed by CCK-8. b. EdU assay for SW480 and HCT116 cells treated with different concentrations (NC, 0.5mg/L, 1mg/L, 2mg/L) of AA, cultured with fatty acid free FBS. c. Representative image of the

subcutaneous tumors and PCNA expression in tumor tissues with AA (2mg/L) treatment via intratumor injection in SW480 and HCT116 cells (n=6 mice per group). Tumor weight and volume (length*width²/2) were calculated. Scale bars, 100 μm.

Fig.2D-G and Fig.S1A-S1C, The effect of AA at increasing the proportion of gram-negative bacteria and decreasing gram-positive bacteria is intriguing, but the exact mechanism is unclear. To further shed some light in this, the authors should measure the microbiome in the faeces of Apc WT mice following the addition of AA. Furthermore, the authors should establish what is the effect of antibiotic administration in CRC progression in regular diet fed Apc^{-/-} mice without the supplementation of AA. An experiment combining fecal transplantation and antibiotic treatment in regular diet fed mice would also provide further confidence on the proposed model.

Response:

Thanks for the reviewer's kindly suggestion.

We measure the microbiome in the faeces of Apc WT mice with or without the addition of AA, and the results also showed AA increased the proportion of gram-negative bacteria and decreasing gram-positive bacteria in Apc WT mice (as shown in Supplementary Fig.1d, e).

An experiment combining fecal transplantation and antibiotic treatment in regular diet fed mice was conducted and the results showed: 1. Antibiotic administration had no obvious effects in CRC progression in regular diet fed AOM/DSS mice (one of the reviewers suggest AOM/DSS model should be applied instead of the AOM model. We repeated the related experiments by using AOM/DSS model in this study) and Apc^{-/-} mice. 2. Stool-AA obviously promoted CRC progress compared with stool-NC, and this effect was not due to antibiotic treatment. 3. Removing gram-negative bacteria by aztreonam completely eliminated the tumor-promoting effect of AA and removing gram-positive bacteria by vancomycin had no effect on tumor-promoting role of AA (as shown in Fig 2h-k, Supplementary Fig 2f-i).

In fact, in this study, for the use of antibiotics, we did it before the actual experiment started to avoid the interference of antibiotics on tumors, this approach has been used in other studies^(2, 3) (line 459-463, 493-499 in Supplementary material).

Supplementary Fig.1d, e: d. Ratio of gram-negative bacteria to gram-positive bacteria in the stool of Apc WT mice (n = 6). e. Relative abundance of *Es. coli*, *En. faecalis* and *Lactobacillus* in the stool of Apc WT mice (n = 6).

Fig 2h-k: h. Representative images of AOM/DSS-induced adenoma with AA feeding under different antibiotics treatment (Quadruple-antibiotics, aztreonam, vancomycin) (n = 6). i. Tumor number of AOM/DSS-induced adenoma with AA feeding under different antibiotics treatment. j. Representative images of HE staining and PCNA expression in AOM-induced adenoma with AA feeding under different antibiotics treatment. *** $p < 0.001$ (compared with NC group). k. Tumor grades of AOM/DSS-induced adenoma with AA feeding under different antibiotics treatment.

*** $p < 0.001$ (compared with NC group)

Supplementary Fig 2f-h: f. Representative tumor images of the intestine-specific *Apc*^{-/-} mice (n = 6). g. The tumor number of the intestine-specific *Apc*^{-/-} mice (n = 6). *** $p < 0.001$ (compared with NC group). h. Representative images of HE staining and PCNA expression in the tumor tissue of the intestine-specific *Apc*^{-/-} mice. i. The tumor grades of the intestine-specific *Apc*^{-/-} mice. *** $p < 0.001$ (compared with NC group)

Figure 3: There are several enzymes contributing to AA metabolism including the PLA enzymes that regulate its release from phospholipids, yet the gene expression analysis presented in Fig.3A is only performed in the genes that are involved in AA synthesis. The gene expression analysis in GDS4382 should be extended to all the genes that play a role in AA metabolism and the results should be validated in the GSE41657 dataset presented in Fig 7.

Response:

Thanks for the reviewer's valuable suggestion. We have analyzed PLA enzymes that play a role in AA metabolism in GDS4382 (Supplementary Fig 3) and GSE41657 (Supplementary Fig 9). The results showed that, almost all PLA enzymes had no expression difference in NC and CRC samples. And no PLA enzymes were obviously

upregulated in CRC samples, compared with NC samples.

Supplementary Fig 3

Supplementary Fig 9

Supplementary Figure 3. The mRNA expression of PLA enzymes in GDS4382. (a-s) The mRNA expression of *PLA2G7*, *PFAH2*, *PFAH1B1*, *PFAH1B2*, *PLA2G4A*, *PLA2G4C*, *PLA2G4D*, *PLA2G6*, *PNPLA2*, *PNPLA4*, *PNPLA5*, *PLA2G1B*, *PLA2G2A*, *PLA2G2D*, *PLA2G2E*, *PLA2G2F*, *PLA2G5*, *PLA2G12A*, *PLA2G12B* in NC and CRC samples.

Supplementary Figure 9. The mRNA expression of PLA enzymes in GSE41657. (a-s) The mRNA expression of *PLA2G7*, *PFAH2*, *PFAH1B1*, *PFAH1B2*, *PLA2G4A*, *PLA2G4C*, *PLA2G4D*, *PLA2G6*, *PNPLA2*, *PNPLA4*, *PNPLA5*, *PLA2G1B*, *PLA2G2A*, *PLA2G2D*, *PLA2G2E*, *PLA2G2F*, *PLA2G5*, *PLA2G12A*, *PLA2G12B* in NC, Adenoma and CRC samples.

Figure 4: It is intriguing that the tumour suppressive effects of *FADS1* appear to be tissue specific. Suppression of *FADS1* induces ROS generation, and it would be useful to check if the effects of *FADS1* knockdown in the microbiome are driven by an increase in redox stress, and if so, whether they can be rescued by treatment with antioxidants. To further establish the cell autonomous vs non-cell autonomous role of AA metabolism in CRC growth it would be useful to assess if AA affects

tumour growth of subcutaneous tumours and whether LPS treatment on its own increases cell proliferation in vitro and subcutaneous tumour growth in vivo.

Response:

Thank you very much for your careful review of our manuscript. Your suggestion is very helpful for us to further explain the mechanism of FADS1-AA axis in regulating microbiome. According to your suggestion, we carried out relevant experiments, and the results showed that FADS1 KD was not increased, but obviously decreased the ROS level in tumor tissues, which was contrary to Luyan Tang's research⁽⁴⁾ (response to reviewer Figure 2a). This may be due to the different role of FADS1 in different cancer types. Generally, dihomo- γ -linolenic acid (DGLA) exhibits potent anti-inflammatory activities, which contrast with its downstream product, arachidonic acid (AA). DGLA treatment obviously inhibited ROS level in the RAW264.7 cells upon LPS activation⁽⁵⁾. AA provoked significant production of ROS in hepatic cancer cells⁽⁶⁾. NAC, an antioxidant, had no significant effect on microbiome (response to reviewer Figure 2b, c). These results indicated that effects of FADS1 in the microbiome were not driven by redox stress. According to our speculation, ROS level alteration in CRC tumor in our study was due to the microbiome alteration. Numerous studies have been reported that intestinal flora or LPS activation can induce ROS production⁽⁷⁻⁹⁾.

response to reviewer Figure 2: a. ROS levels in orthotopic model established with SW480 and HCT-116 cells; b. Ratio of gram-negative and gram-positive bacteria in orthotopic tumors of BALB/C null mice in sh-NC, sh-FADS1, sh-FADS1+NAC groups; c. Relative abundance of *Escherichia coli*, *Enterococcus faecalis* and *Lactobacillus* in the orthotopic tumor.

It is well known that polyunsaturated fatty acids have significant regulatory effects on intestinal flora⁽¹⁰⁾. One study showed that Eicosapentaenoic acid (EPA) feeding significantly changed the intestinal flora of mice, especially increased the number of lactobacillus, and showed tumor suppressive effect⁽¹¹⁾. Therefore, in our study, as a rate-limiting enzyme of AA synthesis, **we hold the opinion that FADS1 modulated the intestinal microecology of gram-negative by regulating the level of AA in tumor microenvironment, and the high AA microenvironment promoted the formation of microecology of enriched gram-negative, indicating a co-evolution of tumor and intestinal microecology.** Firstly, AA feeding obviously altered the intestinal flora of mice, which was quite similar to the effect of FADS1(Fig 2g, Supplementary Fig 2a-c and Fig 4d, e). FADS1 mediates AA synthesis. AA is stored in cell membranes of resting somatic cells in the form of phospholipids and is released into the cytoplasm in response to various stimuli and become free AA. In the orthotopic xenograft, the AA

levels was significantly decreased in the interstitial fluid of tumor tissues in the shFADS1 group, compared with the shNC group (Supplementary Fig.4h, 5f), indicating FADS1 played a role in creating a high AA microenvironment. Moreover, elimination of gut microbes with quadruple-antibiotic administration, the AA showed low levels and no statistical difference in the interstitial fluid of tumor tissues in the shFADS1 group and shNC group (Supplementary Fig.4h, 5f), indicating gut microbes also involved the formation of high AA microenvironment. *In vitro* experiment, under the stimulation of LPS, FADS1 KD showed a lower level of AA in culture supernatant than shNC. (Fig 3f-h). These results suggested that **high AA microenvironment and enrich gram-negative microecology was a mutually reinforcing process.** Furthermore, we used aztreonam to eliminate gram-negative bacteria and vancomycin to eliminate gram-positive bacteria, and the aztreonam treatment showed a decreased AA level, whereas the vancomycin treatment had a roughly equivalent AA level compared with non-antibiotic treatment in the interstitial fluid of shNC tumor tissues in the orthotopic xenograft (Supplementary Fig.4i, 5f). Therefore, **FADS1-AA axis contributed to the creation of a high AA microenvironment in CRC under intestinal flora activation, and in turn became the foundation for enrich gram-negative microecology.**

Fig 2

Fig 2g: Relative abundance of *Es. coli*, *En. faecalis* and *Lactobacillus* in the stool and tumor tissue of C57BL6J mice, NC: normal feeding, n = 8, AA: arachidonic acid feeding, n = 8.

Supplementary Fig 2a-c: a. Ratio of gram-negative bacteria and gram-positive bacteria in the stool of AOM/DSS induced mice, NC: normal feeding, AA: arachidonic acid feeding. b. Ratio of gram-negative bacteria and gram-positive bacteria in the stool of intestinal specific *Apc*^{-/-} mice, NC: normal feeding, AA: arachidonic acid feeding. c. Relative abundance of *Es. coli*, *En. faecalis* and *Lactobacillus* in the stool and tumor tissue of intestinal specific *Apc*^{-/-} mice, NC: normal feeding, AA: arachidonic acid feeding, n = 6.

Fig 4d, e: d. Ratio of gram-negative and gram-positive bacteria in orthotopic tumors of BALB/C null mice injected with shNC and shFADS1 cells. e. Relative abundance of *Escherichia coli*, *Enterococcus faecalis* and *Lactobacillus* in the orthotopic tumor.

Supplementary Fig.4

Supplementary Fig.5

Supplementary Fig.4h,i: h. AA levels in the interstitial fluid of shNC, shFADS1, shNC+quadruple-antibiotic and shFADS1+quadruple-antibiotic orthotopic tumors (n=6). i. AA levels in the interstitial fluid of shNC, shFADS1, shNC+aztreonam, shNC+vancomycin orthotopic tumors (n=6).

Supplementary Fig.5f: AA levels in the interstitial fluid of shNC, shFads1, shFADS1, shNC+quadruple-antibiotic, shFads1+quadruple-antibiotic, shNC+aztreonam, shNC+vancomycin orthotopic tumors established by organoid model (n=6).

Fig 3f-h: f. AA levels in culture medium from SW480 and HCT116 cells transfected with shNC or shFADS1, under activation with 0.5µg/ml and 1µg/ml of LPS, respectively; g. EPA levels in culture medium of SW480 and HCT116 cells transfected with shNC or shFADS1, under activation with 0.5µg/ml and 1µg/ml of LPS, respectively. h. AA/EPA ratio in culture medium of SW480 and HCT116 cells transfected with shNC or shFADS1.

We further used subcutaneous tumor experiment and AA treatment by intratumor injection, and the results showed that AA did not affect the growth of subcutaneous tumor in mice (Supplementary Fig 1c).

In vitro and vivo experiment, LPS activation promoted cell proliferation in shNC CRC cells, but inhibits cell proliferation to some extent in shFADS1 CRC cells (Supplementary Fig 6a-e). Therefore, LPS activation showed drastically different results in shNC and shFADS1 CRC cells, indicating that LPS motivated a certain aspect of FADS1 function.

Supplementary Fig 1c: Representative image of the subcutaneous tumors and PCNA expression in tumor tissues treated by AA in SW480 and HCT116 cells (n = 6), tumor weight and volume (length*width²/2) were calculated.

Supplementary Figure 6. a. The viability of SW480 and HCT116 cells transfected with shFADS1 or shNC, under activation of 1µg/ml of LPS and 2mg/L of AA treatment, as analyzed with CCK-8 assay. b. Colony formation ability of SW480 and HCT116 cells transfected with shFADS1 or shNC, under activation of 1µg/ml of LPS and 2mg/L of AA treatment, as analyzed with colony formation

assay. *** $p < 0.001$ (compared with shNC). c. The viability of SW480 and HCT116 cells transfected with shFADS1 or shNC, under activation of $1\mu\text{g/ml}$ of LPS and 2mg/L of AA treatment, cultured with fatty acid free FBS. d. Colony formation ability of SW480 and HCT116 cells transfected with shFADS1 or shNC, under activation of $1\mu\text{g/ml}$ of LPS and 2mg/L of AA treatment, cultured with fatty acid free FBS. *** $p < 0.001$ (compared with shNC). e. Representative image of the subcutaneous tumors injected with shNC and shFADS1 SW480 or HCT116 cells, under activation of $1\mu\text{g/ml}$ of LPS and 2mg/L of AA or PGE2 treatment by intratumoral injection ($n = 6$). *** $p < 0.001$ (compared with shNC).

Fig.4F-G: To highlight the greater impact of microbiome removal in FADS1 KD cells, it would be useful to plot the changes in tumour growth that can be attributed to the treatment with antibiotics only. Comparing CT imaging source intensity or PCNA staining between experiments 4B-C and 4F-G would provide more confidence that the effect is not only due to the antibiotic treatment. It would also be useful to report if any effect on mouse weight was observed.

Response:

Thank you very much for your careful review of our manuscript. The effect of antibiotic treatment itself on colorectal cancer tumor growth is critical to the accuracy of the conclusions of this study. In previous studies, the role of antibiotics in colorectal cancer has not been clearly determined, and there is no clear evidence that antibiotics have a definite inhibitory effect on cancer. During human's lifetime, because of the need to resist inflammation and infection, use of antibiotics is very common. But whether the use of antibiotics has the function for the development of tumor is still non-consensus, this may be due to the time point, type, dosage, and duration of antibiotic use were quite inconsistent and personalized. **In this study, we did the antibiotics treatment during 2 weeks before the actual experiment started to avoid the interference of antibiotics on tumors (shown in Supplementary materials:Animal experiments).**

In this study, experiments 4B-C and 4F-G were conducted in the same batch (Same time, same conditions). We divided the results into two parts because of the need of logical presentation during article writing, which may have caused some

misunderstandings. Therefore, we integrate 4B-C and 4F-G together in Fig 4b, c and the difference in tumor size between the sh-NC group and the sh-FADS1 group disappeared after the use of antibiotics. Moreover, the compared results also showed that there was no significant additional effect of antibiotic treatment on tumor (when sh-FADS1 and sh-FADS1+antibiotics groups were compared), indicating that the effect of antibiotics itself on tumor was not the decisive factor, while the change of intestinal flora caused the difference (Fig 4b, c).

To further illustrate this point, we designed a selective antibiotic experiment. We used aztreonam to eliminate gram-negative bacteria and vancomycin to eliminate gram-positive bacteria, the results showed aztreonam treatment obviously inhibited the tumor growth and vancomycin treatment had no significant inhibited role in the tumor growth in sh-NC mice (Fig 4f, Supplementary Fig 4f). These results further indicated that FADS1 promoted tumor growth by increasing the proportion of gram-negative bacteria. Finally, we found no significant weight differences among groups of mice (response to reviewer Figure 3a, b).

Fig 4b, c: b. CT imaging of orthotopic tumor in the cecum of BALB/C null mice injected with shNC and shFADS1 SW480^{Luc} cells (n = 6, the red signal represent tumors). c. CT imaging of orthotopic tumors in the cecum of BALB/C null mice

injected with shNC and shFADS1 HCT116^{Luc} cells (n = 6, the red signal represent tumors).

Fig 4f: IVIS imaging of orthotopic tumor in the cecum of BALB/C null mice injected by shNC and shFADS1 SW480^{Luc} cells (n = 6), gut microbes were selectively deleted by aztreonam or vancomycin treatment for 2 weeks before the actual experiment started.

Supplementary Fig 4f: IVIS imaging of orthotopic tumor in the cecum of BALB/C null mice injected by shNC and shFADS1 HCT-116^{Luc} cells (n = 6), gut microbes were selectively deleted by aztreonam or vancomycin treatment for 2 weeks before the actual experiment started.

response to reviewer Figure 3: a. Mice weight burdened with sh-NC, sh-FADS1, sh-NC+antibiotics, sh-FADS1+antibiotics SW480 cells; b. Mice weight burdened with sh-NC, sh-FADS1, sh-NC+antibiotics, sh-FADS1+antibiotics HCT116 cells.

Fig.S3A-B: FADS1 KD appears to decrease cell proliferation upon LPS activation. However, by comparison with tumour weight and volume in Fig.S2D-E, it seems that LPS leads to an increase in tumour volume and weight in shNC condition rather than causing a decrease in FADS1 KD (providing that the duration of experiments is similar, which is not mentioned). It would be useful to repeat the experiments in S3A and S3B in fatty acid free conditions and assess if addition of AA can rescue the decrease in cell proliferation that is observed in FADS1 KD cells in the presence of LPS treatment.

Response:

Thank you very much for your careful review of our manuscript. The experiments in Figure S2 and Figure S3 were not carried out at the same time. Since we found that FADS1 KD did not affect cell proliferation under normal culture *in vivo and in vitro*, we later verified the results under LPS stimulation. Therefore, it is not appropriate to compare FigS2 and FigS3 together. In order to further clarify this point, we repeated these experiments combined with or without LPS activation *in vivo and in vitro* in Supplementary Fig 6. The results showed that compared with no LPS treatment, the cell proliferation ability of shNC cells was significantly increased under LPS stimulation, but the cell proliferation ability of shFADS1 cells was decreased to some extent, indicating FADS1 showed a tumor-promoting role under LPS treatment (LPS treatment simulated intestinal microbiota stimulation) (Supplementary Fig 6a-d), which was consistent to *in vivo* experiments (Supplementary Fig 6e). We also repeated the experiments in S3A and S3B in fatty acid free conditions and the results showed AA rescued the decrease in cell proliferation that was observed in FADS1 KD cells in the presence of LPS treatment (Supplementary Fig 6c-e).

Supplementary Figure 6. a. The viability of SW480 and HCT116 cells transfected with shFADS1 or shNC, under activation of 1µg/ml of LPS and 2mg/L of AA treatment, as analyzed with CCK-8 assay. b. Colony formation ability of SW480 and HCT116 cells transfected with shFADS1 or shNC, under activation of 1µg/ml of LPS and 2mg/L of AA treatment, as analyzed with colony formation assay. ***p<0.001 (compared with shNC). c. The viability of SW480 and HCT116 cells transfected with shFADS1 or shNC, under activation of 1µg/ml of LPS and 2mg/L of AA treatment, cultured with fatty acid free FBS. d. Colony formation ability of SW480 and HCT116 cells transfected with

shFADS1 or shNC, under activation of 1µg/ml of LPS and 2mg/L of AA treatment, cultured with fatty acid free FBS. ***p<0.001 (compared with shNC). e. Representative image of the subcutaneous tumors injected with shNC and shFADS1 SW480 or HCT116 cells, under activation of 1µg/ml of LPS and 2mg/L of AA or PGE2 treatment by intratumoral injection (n = 6). ***p<0.001 (compared with shNC).

Figure 5: The authors should suggest at least an explanation as to why FADS1 KD in vitro results in a decrease in AA but no changes in PGE2, and what is the proposed mechanism of decreased PTGS and PTGES expression following FADS1 KD.

Response:

Thanks for the reviewer's kind suggestion. According to our experimental findings, FADS1 KD significantly down-regulates the synthesis of total AA in CRC cells, but had no effect on free AA level *in vitro* (Fig. 3d, f). AA is stored in cell membranes of resting somatic cells in the form of phospholipids and is released into the cytoplasm in response to various stimuli and become free AA. Only free AA can exert its physiological effects and metabolize to downstream products, like PGE2. Moreover, we also found that FADS1 KD has no effect on the expression of PTGS2 and PTGES, and PGE2 level (Fig. 5e, i). These results indicated that FADS1 was only responsible for regulating the synthesis of AA, and could not directly regulate the metabolism of AA into PGE2. However, the expression of PTGS2 and PTGES in CRC cells was significantly increased by LPS stimulation (Simulated enriched gram-negative bacteria) or fecal transplants from AA-fed mice (Fig. 5g, j), which explains the different phenotype of FADS1 role *in vitro* or *in vivo*. In brief, FADS1 was only responsible for regulating the synthesis of AA. Without gut microbes, FADS1 did not show a cancer-promoting phenotype because FADS1 could not directly activate the metabolic pathway of AA. When gut microbiota is present, FADS1-AA axis creates an intestinal microecology of enriched gram-negative bacteria. The enriched gram-negative bacteria induced the expression of PTGS2 and PTGES, and promoted the PGE2 synthesis via

TLR4 in CRC cells (We also addressed this issue in the Discussion section, line 390, 414-416, 421-425 in the revised manuscript). And a study showed that the TLR4/MYD88-NF- κ B pathway was involved in PTGS2 gene transcription that promoted the conversion of AA to PGE2⁽¹²⁾. PTGES was also transcriptional regulated by NF- κ B⁽¹³⁾. Therefore, FADS1 indirectly regulated PTGS2 and PTGES expression by creating an intestinal microecology of enriched gram-negative bacteria.

Figure 6: Fig.6F To attribute the increase in cell proliferation solely to the increase of AA-mediated production of PGE2, it would be interesting to see if addition of PGE2 can rescue TLR4/MYD88 KD-mediated inhibition of cell viability after LPS treatment in SW480 and HCT116 in vitro. The authors should also check if inhibition of the TLR4/MYD88 signalling pathway in vivo rescues the effect of AA addition at increasing tumour growth.

Response:

Thanks for the reviewer's good suggestion. As suggested by the reviewer, we have used PGE2 to conduct rescue assay and the results showed adding PGE2 rescued TLR4/MYD88 KD-mediated inhibition of cell viability after LPS treatment in SW480 and HCT116 in vitro (Fig 6f). In AOM-DSS and *Apc*^{-/-} mice, inhibition of the TLR4/MYD88 signalling pathway in vivo by TLR4 inhibitor (TAK-242) and PTGS2 inhibitor (Celecoxib) rescues the effect of AA addition in increasing tumour growth (Supplementary Fig 8d, e).

Fig 6f: The viability of SW480 and HCT116 cells transfected with siTLR4, siMYD88, siPTGES,

siPTGS2 and siNC activated by 1 μ g/ml LPS.

Supplementary Fig 8d, e: d. Representative image of AOM/DSS model with NC and AA feeding, treated by TLR4 inhibitor (TAK-242) and PTGS2 inhibitor (Celecoxib); e. Representative image of *Apc*^{-/-} model with NC and AA feeding, treated by TLR4 inhibitor (TAK-242) and PTGS2 inhibitor (Celecoxib).

Minor comments:

Figure 1A: It is not clear how was Arachidonic Acid (AA) levels measured? The authors should provide more information in the materials and method section on how AA was extracted from samples, whether these are frozen or paraffine embedded etc.

Response: Thanks for the reviewer's professional suggestions. AA levels were measured by detection kit (NBP2-59872, Novus Biologicals). For sample preparation, fresh tumor tissue was used, cut, and weighed 100mg, frozen quickly with liquid nitrogen, ground the tissue manually, added 1ml PBS, and homogenized further with ultrasonic cell crusher. In vitro experiments, for total AA, 10⁷ CRC cells were used and digested, 0.2ml PBS was added and repeated freeze-thaw with liquid nitrogen, and homogenized further with ultrasonic cell crusher. All the samples were centrifuged for 10 minutes (3000 rpm) and supernatant were collected. According to instructions,

detection procedure included adding standard samples, adding test samples, adding enzymes, incubation at 37°C for 60 min, washing, coloring, stopping the reaction and measurement (shown in Supplementary materials: Elisa assay).

Figure 1F: mentions tumour grade but this was not assessed for the APC^{-/-} model following AA or NC diet.

Response: Thanks for the reviewer's valuable suggestions. We had assessed the tumour grade for the *Apc^{-/-}* model following AA or NC diet and the results indicated that AA feeding showed a higher tumor grade in *Apc^{-/-}* model, which showed in Fig 1j.

Fig 1j: Tumor grade of intestinal specific *Apc^{-/-}* mice.

Figure 2F: The histogram represents distribution for NC=8 and AA=8, however the Figure legend for 2c mentions n=6 for both groups. If this is from the same experiment, the number of mice per group need to be revised.

Response: Thanks for the reviewer's kindly suggestion. We are extremely sorry about a writing error in Figure legend 2C. It should be n=8 in NC and AA groups. We had revised it in Figure legend 2c.

Figure 2J and supplementary figure 2F are exactly the same figure (Tumor number per mice). Was the number of tumours identical in both experiments with the two different models or is this a mistake when

compiling the figures?

Response: Thank you very much for your careful review of our manuscript. Supplementary figure 2F was careless mistake and showed a repeat picture (repeat with Figure 2F, Fig 2i in revised Fig 2). We had revised it in Supplementary Fig 2g.

Fig 2i: tumor number of AOM/DSS-induced mice (n = 6).

Supplementary Fig 2g: tumor number of *Apc*^{-/-} mice (n = 6).

Figure 3B: legend in text line 252 mentions AA but the figure is about EPA. AA increase was in figure 1A. Text or figure legend should be corrected.

Response: Thanks for the reviewer's careful review of our manuscript. We have carefully checked and corrected relevant errors. We revised as follows: AA level was gradually upregulated in CRC process (Fig.1a) and positively correlated with FADS1 expression (Fig.3c). However, EPA level showed no significant change and was not correlated with FADS1 expression (Fig.3b, c) (line 162-166 in the revised manuscript).

Line 286: observations are made for figure 4F and G but only 4G is mentioned in the text.

Response: Thank you very much for your careful review of our manuscript. We had

changed it in the manuscript (line 178-180 in the revised manuscript). We integrate 4B-C and 4F-G together in Fig 4b, c.

Fig 4b, c: Computed tomography (CT) with 3D organ reconstruction bioluminescence imaging of the orthotopic tumor of BALB/C null mice injected with shNC and shFADS1 SW480^{Luc} and HCT116^{Luc} cells (n=6 mice per group). Gut microbes were deleted by quadruple antibiotics for two weeks before the experiment started. The yellow signal represents intestine, the green signal represents colon and the red signal represents tumor.

Line 719: microbeq2-mediated in figure legend.

Response: Thanks for the reviewer's professional suggestions. We had revised it in the manuscript. We revised as follows: Figure 6. The TLR4/MYD88 pathway is involved in gut microbe-mediated prostaglandin E2 production (line 939-940 in the revised manuscript).

Line 430: the authors wrote that FADS1 KD induces significantly reduced proliferation of CRC cells in vivo and in vitro. They then write in line 434 that FADS1 KD does not show significant inhibition of cell proliferation. This should be corrected.

Response: Thank you for the reviewer's reminding. We apologize for the confusion in this section. According to our study, we found FADS1 KD induces significantly reduced

proliferation of CRC cells *in vivo*, but not *in vitro*. With the LPS treatment *in vitro*, the proliferation ability of CRC cells showed a significant difference in shFADS1 and shNC groups. These indicated that FADS1 promotes tumor growth by a mechanism involving a collaborative intracellular pathway mediated by intestinal microbes and FADS1. In line 434, it is a reference's research results, in which, FADS1 knockdown also did not affect cell proliferation in CRC and breast cancer cells *in vitro*. We had revised it in the manuscript (line 382-384, line 409-414 in the revised manuscript).

Finally, this manuscript would benefit from more detailed explanation of the results, and some editing help from someone with proficiency in English and scientific writing.

Response: We have added more description in the results in the revised manuscript. And we invited a native editor to revised our article in grammar and writing.

Reviewer #2 (Remarks to the Author):

In the manuscript "The FADS1/AA axis enhances AA metabolism by altering intestinal microecology in CRC" Xu et al. report a finding that AA increase/feeding positively influences CRC growth. The manuscript is potentially interesting, however, the mechanistic data is rather weak and mostly based on in vitro cell line experiments. The authors overinterpret some of the findings based on inadequate in vivo models.

- the data on the shift to gram neg bacteria following AA feeding is interesting. The authors fail to prove that in in vivo experiments. The use of antibiotic treatment generally abolishes CRC growth - therefore proving the point the reviewers make that the AA effects is mediated by gram neg bacteria in this set up is not possible. This set up is used in Fig 2 and 4 and not valid to answer this effectively

- for in vivo experiments the authors use immunocompromised mice and human cells lines. This set up does not justify the claim of the authors. Organoid models using immunocompetent mice need to be used.

- showing that prostaglandin E2 levels depend on LPS activation of TLR4 is a nice observation in vitro. However, this is not proven in vivo and therefore the data is overinterpreted.

Comment 1:

- the data on the shift to gram neg bacteria following AA feeding is interesting. The authors fail to prove that in in vivo experiments. The use of antibiotic treatment generally abolishes CRC growth - therefore proving the point the reviewers make that the AA effects is mediated by gram neg bacteria in this set up is not possible. This set up is used in Fig 2 and 4 and not valid to answer this effectively

Response:

Thank you very much for your careful review of our manuscript. The effect of antibiotic treatment itself on CRC tumor growth is critical to the accuracy of the conclusions of this study. In previous studies, the role of antibiotics in colorectal cancer has not been clearly determined, and there is no clear evidence that antibiotics have a definite

inhibitory effect on cancer. During human's lifetime, because of the need to resist inflammation and infection, use of antibiotics is quite common. But whether the use of antibiotics has the function for the development of tumor is still non-consensus, this may be due to the time point, type, dosage, and duration of antibiotic use were quite inconsistent and personalized. **In this study, we did the treatment of antibiotics during 2 weeks before the actual experiment started to avoid the interference of antibiotics on tumors, this approach has been used in other studies^(2, 3).**

In view of your good suggestions, we adjusted the style of data presentation in the Figure and designed new experiments, which maybe answer your question to some extent.

For Fig 2: We did not mean that **the use of antibiotic treatment generally abolishes CRC growth.** In our study, we found AA feeding shifted to gram neg bacteria. **Pretreated with quadruple antibiotics to delete intestinal flora (quadruple antibiotics were used before starting the experiment in the AOM/DSS and *Apc*^{-/-} model) abolishes AA effect on CRC tumor.** Fecal transplantation of AA feeding could promote growth of CRC tumor. To make a better point, a repeat experiment combining fecal transplantation and antibiotic treatment was conducted and the results showed: 1. Antibiotic administration had no obviously effect in CRC progression in regular diet fed AOM/DSS mice (**one of the reviewers suggest AOM/DSS model should be applied instead of the AOM model. We repeated the related experiments in this study**) and *Apc*^{-/-} mice. 2. Stool-AA obviously promoted tumor progress compared with stool-NC, and this effect was not due to antibiotic treatment. 3. Removing gram-negative bacteria by aztreonam completely eliminated the tumor-promoting effect of AA and removing gram-positive bacteria by vancomycin had no effect on tumor-promoting role of AA (as shown in **Fig.2h-k and Supplementary Fig.2f-i**). Moreover, further analysis revealed that stool-AA and AA feeding+vancomycin groups both showed a similar gut microbiota composition with AA feeding groups, and stool-NC and AA feeding+aztreonam groups both showed a similar gut microbiota composition with NC feeding groups (as shown in **Supplementary Fig.2j**).

For Fig 4, experiments 4B-C and 4F-G were conducted in the same batch (Same time,

same conditions). We divided the results into two parts because of the need of logical presentation during article writing, which may have caused some misunderstandings. So, we integrated 4B-C and 4F-G together in Fig 4b, c, and the difference in tumor size between the sh-NC group and the sh-FADS1 group disappeared upon the use of antibiotics. Moreover, the compared results also showed that there was no significant additional effect of antibiotic treatment on tumor (if sh-FADS1 and sh-FADS1+antibiotics groups were compared), indicating that the effect of antibiotics itself on tumor was not the decisive factor, but the change of intestinal flora caused the difference (Fig 4b, c).

To further clarify AA effects were mediated by gram-negative bacteria, we designed a selective antibiotic experiment. We used aztreonam to eliminate gram-negative bacteria and vancomycin to eliminate gram-positive bacteria, the results showed aztreonam treatment obviously inhibited the tumor growth and vancomycin treatment had no significant inhibited role in the tumor growth in sh-NC mice (Fig 4f, Supplementary Fig 4f). Further analysis revealed that shNC+vancomycin groups showed a similar gut microbiota composition with shNC groups, and shNC+aztreonam groups showed a similar gut microbiota composition with shFADS1 groups (Supplementary Fig.4g). These results further indicated that FADS1-AA axis promoted CRC tumor growth by modulating the intestinal microecology of gram-negative.

Fig 2h-k: h. Representative images of AOM/DSS-induced adenoma with AA feeding under different antibiotics treatment (Quadruple-antibiotics, aztreonam, vancomycin) (n = 6). i. Tumor number of AOM/DSS-induced adenoma with AA feeding under different antibiotics treatment. j. Representative images of HE staining and PCNA expression in AOM-induced adenoma with AA feeding under different antibiotics treatment. k. Tumor grades of AOM/DSS-induced adenoma with AA feeding under different antibiotics treatment.

Supplementary Fig 2f-j: f. Representative tumor images of the intestine-specific *Apc*^{-/-} mice (n = 6). g. The tumor number of the intestine-specific *Apc*^{-/-} mice (n = 6). h. Representative images of

HE staining and PCNA expression in the tumor tissue of the intestine-specific *Apc*^{-/-} mice. i. The tumor grades of the intestine-specific *Apc*^{-/-} mice. j. Gut microbes' alteration in the fecal transplantation and selective antibiotic experiments.

Fig 4b, c: Computed tomography (CT) with 3D organ reconstruction bioluminescence imaging of the orthotopic tumor of BALB/C null mice injected with shNC and shFADS1 SW480^{Luc} and HCT116^{Luc} cells (n=6 mice per group). Gut microbes were deleted by quadruple antibiotics for two weeks before the experiment started. The yellow signal represents intestine, the green signal represents colon and the red signal represents tumor.

Fig 4f: IVIS imaging of orthotopic tumor in the cecum of BALB/C null mice injected by shNC and shFADS1 SW480Luc cells (n = 6), gut microbes were selectively deleted by aztreonam or vancomycin treatment for 2 weeks before the experiment started.

Supplementary Fig 4f: IVIS imaging of orthotopic tumor in the cecum of BALB/C null mice injected by shNC and shFADS1 HCT-116^{Luc} cells (n = 6), gut microbes were selectively deleted by aztreonam or vancomycin treatment for 2 weeks before the experiment started.

Supplementary Fig 4g: Ratio of gram-negative and gram-positive bacteria in the feces in the surface of the orthotopic tumors of BALB/C null mice in shNC, shFADS1, shNC+aztreonam, shNC+vancomycin groups. *p<0.05, **p<0.01, ***p<0.001 (compared with shNC)

Comment 2:

- for in vivo experiments the authors use immunocompromised mice and human cells lines. This set up does not justify the claim of the authors. Organoid models using immunocompetent mice need to be used.

Response:

Thanks for the reviewer's kindly suggestion. We constructed a mouse organoid model use AOM/DSS C57BL/6J mouse. Then, we knockdown Fads1 in organoid cells and used those cells of shNC and shFads1 to carry out related experiments *in vivo* and *in vitro*. We found Fads1 significantly inhibited the tumor growth in orthotopic tumor model. However, *in vitro* and in subcutaneous tumor model, Fasd1 knockdown did not affect the cell proliferation of tumor (Supplementary Fig 5). These results were

consistence with the results obtained from the models by immunocompromised mice.

Supplementary Fig 5: a. Establishment of CRC organoid model by AOM/DSS induced mice. b. Transfected organoids with shNC and shFads1; c. IVIS imaging of orthotopic tumor injected by shNC and shFads1 organoid cells (n = 6), gut microbes were deleted by antibiotics treatment for 2 weeks. d. Ratio of gram-negative and gram-positive bacteria in orthotopic tumors of injected with shNC and shFads1 organoid cells. *p<0.05, ***p<0.001 (compared with shNC). e. Representative image of subcutaneous tumors injected by sh-NC and sh-Fads1 organoid cells (n=6), tumor weight and volume (length*width²/2) were calculated. f. AA levels in the interstitial fluid of shNC, shFads1, shFADS1, shNC+quadruple-antibiotic, shFads1+quadruple-antibiotic, shNC+aztreonam, shNC+vancomycin orthotopic tumors (n=6). gut microbes were deleted by antibiotics treatment for 2 weeks before the experiment started.

Comment 3:

- showing that prostaglandin E2 levels depend on LPS activation of TLR4 is a nice observation *in vitro*. However, this is not proven *in vivo* and therefore the data is overinterpreted.

Response:

Thanks for the reviewer's good suggestion. We have supplemented the *in vivo* experiments by subcutaneous tumor model and the results showed inhibition of TLR4/MYD88 by shRNAs obviously decreased prostaglandin E2 levels in subcutaneous tumors with LPS activation (Supplementary Fig 8b, c)

Supplementary Fig 8b, c: b. PGE2 levels in the subcutaneous tumors injected with shTLR4, shMYD88 and shNC SW480 or HCT-116 cells, under activation of 1 μ g/ml of LPS. *** p <0.001. c. Representative image of subcutaneous tumors injected by shTLR4, shMYD88 and sh-NC SW480 or HCT-116 cells, under activation of 1 μ g/ml LPS by intratumoral injection (n = 6).

Reviewer #3 (Remarks to the Author):

In this manuscript, the role of the FADS1/AA axis in colorectal cancer (CRC) via modulating gut microbiota was explored. FADS1 knockdown decreased the ratio of Gram-negative and Gram-positive bacteria which impacted AA metabolism and decreased the production of a primary AA derivative prostaglandin E2 (PGE2) through downregulation of PTGS2 and PTGES. However, PGE2 is known, through numerous studies in humans and mice, to promote colon carcinogenesis. This is the basis for the anti-colon cancer effects of NSAIDs like sulindac. The gut microbiota also influences colon cancer in several experimental models. Thus, the novelty of this study is lacking. The authors need to determine the mechanism by which the gut microbiota changes and how this change affects PGE2.

Comments:

1. FADS1 and FADS2 genes were found highly expressed in colon tumor tissue compared with normal colon tissue in a previous genome-wide association study of CRC in East Asians (PMID: 24836286). However, FADS2 expression showed no change in the present study. Explain the discrepancy?

Response: Thanks for the reviewer's professional suggestions. The reviewer is absolute right, FADS1 and FADS2 was both risk loci in colorectal cancer in a previous genome-wide association study of CRC in East Asians⁽¹⁴⁾. However, in subsequent analysis, we found that FADS2 expression was not significantly elevated in CRC samples in GDS4382, GSE110223 and GSE41657 datasets (Fig 3a, response to reviewer Figure 4a-c). Survival analysis were also found that FADS2 expression did not affect the survival of colorectal cancer patients (response to reviewer Figure 4d). Therefore, Therefore, we investigated the role of FADS1 in CRC as FADS1 would be a more valuable gene in CRC.

Fig 3a: The mRNA expression of *FADS2* in 17 cases of paired NC and CRC samples from GDS4382, NC: normal colon epithelial tissue, CRC: colorectal cancer.

response to reviewer Figure 4: a and b: the mRNA expression of *FADS1* and *FADS2* in NC and CRC samples in GSE110223. c. the mRNA expression of *FADS2* in NC and CRC samples in GSE41657. d. Kaplan–Meier analysis of the relationship between *FADS2* mRNA expression and prognosis of CRC patients from the TCGA dataset, FADS2-low: CRC patients with low mRNA expression of *FADS2* (n = 220), FADS2-high: CRC patients with high mRNA expression of *FADS2* (n = 220).

2. AA was reported to regulate apoptosis. The TUNEL assay should be performed to measure apoptosis in cells and the mouse models.

Response: Thanks for the reviewer’s good suggestion. We conducted TUNEL assay *in vivo* and *in vitro*, with AA treatment. The results showed AA treatment (1mg/L and 2mg/L) did not affect cell apoptosis of CRC both *in vivo* and *in vitro* (response to reviewer Figure 5a, b).

response to reviewer Figure 5: a. Representative images of TUNEL staining in the tumor tissues of the AOM/DSS-induced and intestine-specific *Apc*^{-/-} mice. b. Representative images of TUNEL staining in SW480 and HCT116 cells with 1mg/L or 2mg/L treatment.

3. Since LPS was widely used in cell experiments, the colitis-related CRC murine model (AOM-DSS) should be applied instead of the AOM model.

Response: Thanks for the reviewer's reminding. We have repeated in vivo experiments used AOM-DSS model instead of the AOM model (Fig 1c, Fig 2i, Fig 5g, Supplementary Fig 8d).

4. Images of full-length colons need to be shown instead of trimmed colons.

Response: Thanks for the reviewer's kind suggestion. We showed full-length colons instead of trimmed colons in the revised manuscript.

5. Under LPS treatment, FADS1 knockdown decreased PGE2 production in vitro (Figure 5F). However, expression of PTGES and PTGS2 were not significantly affected at both the mRNA and protein levels in vitro (Figure 5I, S4C-F). Why did PGE2 production decrease while expression of the core synthetic enzymes PTGES and PTGS2 were not changed?

Response: Thanks for the reviewer's kindly suggestion. Under LPS treatment, FADS1 knockdown showed a decreased PGE2 production *in vitro* (Fig 5f). However, without LPS treatment, FADS1 knockdown did not affect PGE2 production *in vitro* (Fig 5e). It was consistent to the expression of PTGES and PTGS2. PTGES and PTGS2 expression was increased under LPS treatment in CRC cells compared to those without LPS treatment *in vitro* (Figure 5i). FADS1 knockdown did not affect the expression of PTGES and PTGS2 *in vitro* (Figure 5i, Supplementary Fig 7e-h). Therefore, FADS1 itself did not directly regulate PTGES and PTGS2 expression and PGE production *in vitro*. *In vivo* experiments, FADS1-AA axis altered intestinal microecology, and the altered intestinal microecology regulated PTGES and PTGS2 expression, and eventually promoted PGE2 production. This explained the difference in FADS1's performance *in vitro* and *in vivo*.

6. Line 212: “.....a spontaneous model of CRC by hybridizing Apc.....” should be “a spontaneous model of CRC by crossing Apc”.

Response: Thanks for the reviewer's kindly suggestion. We had revised it in the manuscript (line 107-109 in the revised manuscript) as follows: Next, a spontaneous model of CRC was established by crossing *Apc^{flx/flx}* and *Lgr5-EGFP-IRES-CreERT2* mice to generate intestine-specific *Apc^{-/-}* mice.

7. Line 236: “Gut microbial elimination by antibiotic administration completely inhibited the tumor-promoting activity of AA (Figure 2H, Figure S1D)”. Tumor numbers should be shown and statistically analyzed.

Response: Thanks for the reviewer's kind suggestion. We have revised it in the manuscript (Fig 2i, Supplementary Fig 2g).

Fig 2h, i: h. Representative images of AOM/DSS-induced adenoma with AA feeding under different antibiotics treatment (Quadruple-antibiotics, aztreonam, vancomycin) (n = 6). i. Tumor number of AOM/DSS-induced adenoma with AA feeding under different antibiotics treatment. ***p<0.001 (compared with NC)

Supplementary Fig 2f, g: f. Representative tumor images of the intestine-specific *Apc*^{-/-} mice (n = 6). g. The tumor number of the intestine-specific *Apc*^{-/-} mice (n = 6). h. Representative images of HE staining and PCNA expression in the tumor tissue of the intestine-specific *Apc*^{-/-} mice. ***p<0.001 (compared with NC)

8. Line 250: “AA and eicosapentaenoic acid (EPA) are two products of FADS1 catalysis. AA level was gradually upregulated in CRC and positively correlated with FADS1 expression (Figure 3B).” This should be Figure 3C. “However, EPA level showed no significant change and was not correlated with FADS1 expression (Figure 3C).” This should be Figure 3B&C.

Response: Thanks for the reviewer’s valuable suggestions. We had revised these errors in the manuscript (line 162-166 in the revised manuscript) as follows: The AA level was gradually upregulated in CRC process (Fig.1a) and positively correlated with the FADS1 expression (Fig.3c). However, the EPA level showed no significant change and was not correlated with the FADS1 expression (Fig.3b, c).

9. Line 261, Figure 3F: “We found that lipopolysaccharide (LPS) activation from *Escherichia coli* significantly increased AA levels in culture supernatant in a concentration-dependent fashion in shNC cells, but not in shFADS1 cells (Figure 3F).” In shFADS1 cells, LPS still increased AA levels, just to a less extent compared to shNC cells. Besides, instead of labeling as “- + +” on x-axis, it should be labeled as “- sh1 sh2”, since two shRNAs were used for each gene. Moreover, the knockdown efficiency of shFADS1, shTLR4, shMDY88, shPTGES, shPTGS2 needs to be determined by qPCR and WB.

Response: Thanks for the reviewer’s valuable suggestions. We had revised and describe the results more accurately in the manuscript (line 228-231 in the manuscript) as follows: Lipopolysaccharide (LPS) activation from *Escherichia coli* showed obviously increased AA level in the culture supernatant in a concentration-dependent manner in shNC cells, but only showed a slight elevation in shFADS1 cells.

We also modified the labels in the figures (Fig 3d-g, Supplementary Fig7i). In addition, knockdown efficiency of shFADS1 was showed in Supplementary Fig 4b, and we calculated the knockdown efficiency of shTLR4, shMDY88, shPTGES, shPTGS2 and showed in Supplementary Fig7i, Supplementary Fig 8a.

Supplementary Fig 4b:FADS1 knockdown in SW480 and HCT116 cells by shRNA.

Supplementary Fig7i: PTGES and PTGS2 knockdown in SW480 and HCT116 cells by siRNA and shRNA.

Supplementary Fig 8a: TLR4 and MYD88 knockdown in SW480 and HCT116 cells by siRNA and shRNA.

10. Line 281: The mechanism by which knockdown of FADS1 reduced Escherichia coli and Enterococcus faecalis, and increased Lactobacillus needs to be determined.

Response: Thanks for the reviewer's valuable suggestions. It is well known that polyunsaturated fatty acids have significant regulatory effects on intestinal flora^(10, 15), one study showed that Eicosapentaenoic acid (EPA, n-3 PUFA) feeding significantly changed the intestinal flora of mice, especially increased the number of lactobacillus and showed tumor suppressive effect⁽¹¹⁾. Therefore, in our study, as a rate-limiting enzyme of arachidonic acid synthesis, we hold the opinion that **FADS1 regulates intestinal flora by regulating the level of AA in tumor microenvironment**. Firstly, AA feeding obviously altered the intestinal flora of mice, which was similar to the effect of FADS1(**Fig 2g, Supplementary Fig 2a-c and Fig 4d, e**). FADS1 mediates AA synthesis. AA is stored in cell membranes of resting somatic cells in the -form of phospholipids and is released into the cytoplasm in response to various stimuli. In the orthotopic xenograft, the AA levels was significantly decreased in the interstitial fluid of tumor tissues in the shFADS1 group, compared with the shNC group (**Supplementary Fig.4h, 5f**), indicating FADS1 played a role in creating a high AA microenvironment. Moreover, elimination of gut microbes with quadruple-antibiotic

administration, the AA showed low levels and no statistical difference in the interstitial fluid of tumor tissues in the shFADS1 group and shNC group (Supplementary Fig.4h, 5f), indicating gut microbes also involved the formation of high AA microenvironment. *In vitro* experiment, under the stimulation of LPS, FADS1 KD significantly reduced the level of AA in culture supernatant (Fig 3f-h). These results suggested that high AA microenvironment and enrich gram-negative microecology was a mutually reinforcing process. Furthermore, we used aztreonam to eliminate gram-negative bacteria and vancomycin to eliminate gram-positive bacteria, and the aztreonam treatment showed a decreased AA level, whereas the vancomycin treatment had a roughly equivalent AA level compared with non-antibiotic treatment in the interstitial fluid of shNC tumor tissues in the orthotopic xenograft (Supplementary Fig.4i, 5f). Therefore, **FADS1-AA axis contributed to the creation of a high AA microenvironment in CRC under intestinal flora activation, and in turn became the foundation for enrich gram-negative microecology**

Fig 2

Supplementary Fig 2

Fig 2g: Relative abundance of *Es. coli*, *En. faecalis* and *Lactobacillus* in the stool and tumor tissue of C57BL6J mice, NC: normal feeding, n = 8, AA: arachidonic acid feeding, n = 8.

Supplementary Fig 2a-c: a. Ratio of gram-negative bacteria and gram-positive bacteria in the stool of AOM/DSS induced mice, NC: normal feeding, AA: arachidonic acid feeding. b. Ratio of gram-negative bacteria and gram-positive bacteria in the stool of intestinal specific *Apc*^{-/-} mice, NC: normal feeding, AA: arachidonic acid feeding. c. Relative abundance of *Es. coli*, *En. faecalis* and *Lactobacillus* in the stool and tumor tissue of intestinal specific *Apc*^{-/-} mice, NC: normal feeding, AA: arachidonic acid feeding, n = 6.

Fig 4d, e: d. Ratio of gram-negative and gram-positive bacteria in orthotopic tumors of BALB/C null mice injected with shNC and shFADS1 cells. e. Relative abundance of *Escherichia coli*, *Enterococcus faecalis* and *Lactobacillus* in the orthotopic tumor.

Supplementary Fig.4

Supplementary Fig.5

Supplementary Fig.4h,i: h. AA levels in the interstitial fluid of shNC, shFADS1, shNC+quadruple-antibiotic and shFADS1+quadruple-antibiotic orthotopic tumors (n=6). i. AA levels in the interstitial fluid of shNC, shFADS1, shNC+aztreonam, shNC+vancomycin orthotopic tumors (n=6).

Supplementary Fig.5f: AA levels in the interstitial fluid of shNC, shFads1, shFADS1, shNC+quadruple-antibiotic, shFads1+quadruple-antibiotic, shNC+aztreonam, shNC+vancomycin orthotopic tumors established by organoid model (n=6).

Fig 3f-h: f. AA levels in culture medium from SW480 and HCT116 cells transfected with shNC or shFADS1, under activation with 0.5µg/ml and 1µg/ml of LPS, respectively; g. EPA levels in culture medium of SW480 and HCT116 cells transfected with shNC or shFADS1, under activation with 0.5µg/ml and 1µg/ml of LPS, respectively. h. AA/EPA ratio in culture medium of SW480 and HCT116 cells transfected with shNC or shFADS1.

11. Line 316, Figure 5I: “FADS1 knockdown did not affect PTGS2 and PTGES expression in vitro”. It looks like PTGES was decreased by FADS1 KD in SW480 cells. All WB bands need to be quantified in this manuscript.

Response: Thanks for the reviewer’s good comments. We repeated this part and the results showed that FADS1 knockdown did not affect PTGS2 and PTGES expression in SW480 and HCT116 cells (Fig 5i). We had quantified all WB bands in this manuscript.

Fig 5i: PTGES and PTGS2 expression in SW480 and HCT116 cells transfected with shFADS1 or shNC activated by 1 μ g/ml of LPS.

12. To prove that FADS1/AA axis affect CRC development via the PTGS2/PTGES-PGE2 pathway, rescue experiments should be performed. For example, 1. overexpress PTGS2/PTGES in shFADS1 cells and measure xenograft growth, 2 supplement PGE2 in shFADS1 cells or mouse models. 3. delete PTGS2/PTGES /inhibit PGE2 pathway in AA treated mice.

Response: Thanks for the reviewer's good comments. We have conducted rescue experiments to prove that FADS1/AA axis affect CRC development via the PTGS2/PTGES-PGE2 pathway.

In detail, supplement PGE2 rescued the cell proliferation in shFADS1 cells under LPS activation (Supplementary Fig 7a,b, Supplementary Fig 6e). knockdown of PTGS2 and PTGES by siRNAs inhibited cell proliferation under LPS activation in CRC cells and supplement PGE2 rescued this phenomenon (Fig 6f). Moreover, in AOM/DSS or *Apc^{-/-}* model, inhibition of PTGS2 by Celecoxib decreased the tumor growth in AA treated mice (Supplementary Fig 8d, e).

Supplementary Fig 7a, b: a. The viability of SW480 cells transfected with shFADS1 or shNC, HCT116

under activation of 1 μ g/ml of LPS and 1mg/L of PGE2 treatment, as analyzed with CCK-8 assay. b. The viability of HCT116 cells transfected with shFADS1 or shNC, under activation of 1 μ g/ml of LPS and 1mg/L of PGE2 treatment, as analyzed with CCK-8 assay.

Supplementary Fig 6e: Representative image of the subcutaneous tumors injected with shNC and shFADS1 SW480 or HCT116 cells, under activation of 1 μg/ml of LPS and 2mg/L of AA or 1 mg/L of PGE2 treatment by intratumoral injection (n = 6). ***p<0.001 (compared with shNC)

Fig 6f: The viability of SW480 and HCT116 cells transfected with siTLR4, siMYD88, siPTGES, siPTGS2 and siNC activated by 1 μg/ml LPS.

Supplementary Fig 8d, e: d. Representative image of AOM/DSS model with NC and AA feeding, treated by TLR4 inhibitor (TAK-242) and PTGS2 inhibitor (Celecoxib); e. Representative image of *Apc*^{-/-} model with NC and AA feeding, treated by TLR4 inhibitor (TAK-242) and PTGS2 inhibitor (Celecoxib).

13. Line 333: “Prostaglandin E2 levels significantly decreased after LPS activation (Figure 6A, 6B)”. Should be “Prostaglandin E2 levels were significantly increased after LPS activation, and decreased by TLR4 or MYD88 KD”.

Response: Thanks for the reviewer’s good comments. We have modified it in the manuscript (line 300-302 in the revised manuscript).

14. What are the mechanisms by which the TLR4/MYD88 pathway regulates the expression of PTGS2 and PTGES? Is the TLR4/MYD88 pathway activated under CRC, or with AA treatment?

Response: Thanks for the reviewer’s valuable suggestions. Extracellular signals activate nuclear factor- κ B (NF- κ B) and mitogen-activated protein kinase cascades, with the assistance of MYD88, leading to transcription of inflammation-related target genes. It was reported that the TLR4/MYD88-NF- κ B pathway is involved in PTGS2 gene transcription that promotes the conversion of AA to prostaglandin E2⁽¹²⁾. PTGES was

also transcriptional regulated by NF- κ B⁽¹³⁾. This is also addressed in the Discussion section in the revised manuscript (line 432-438).

TLR is the primary signal transducer for human intestinal epithelial cells to sense activation by intestinal microorganisms. In our study, TLR4/MYD88 pathway was not directly activated by CRC. TLR4/MYD88 pathway was activated by the altered intestinal microecology by FADS1-AA axis. Therefore, due to FADS1-AA axis, CRC altered the intestinal microecology and the altered intestinal microecology activated TLR4/MYD88 pathway.

15. Transplantation of *Escherichia coli*, *Enterococcus faecalis* or *Lactobacillus* in mice should be carried out in order to determine the role of gut microbiota in CRC.

Response: Thanks for the reviewer’s valuable suggestions. We have conducted related experiments used AOM/DSS model and the results showed *Escherichia coli*, *Enterococcus faecalis* promoted tumor growth, while *Lactobacillus* inhibited tumor growth (response to reviewer Figure 6). These results were consistent with previous reports⁽¹⁶⁻¹⁹⁾.

response to reviewer Figure 6: A. Representative image of AOM/DSS model with *Escherichia coli*, *Enterococcus faecalis* or *Lactobacillus* treatment; B. Tumor number of AOM/DSS model with *Escherichia coli*, *Enterococcus faecalis* or *Lactobacillus* treatment; C. PCNA staining of tumor tissue of AOM/DSS model with *Escherichia coli*, *Enterococcus faecalis* and *Lactobacillus* treatment.

16. To determine the translational potential of this study, a FADS1 inhibitor such as 2-(2,2,3,3,3-pentafluoropropoxy)-3-[4-(2,2,2-trifluoroethoxy) phenyl]-5,7-dihydro-3H-pyrrolo[2,3-d]pyrimidine-4,6-dione (PMID: 31488602), should be applied to xenograft or PDX mice.

Response: Thanks for the reviewer’s valuable suggestions. We had conducted related experiments used xenograft model and the results showed FADS1 inhibitor (D5D-IN-326) obviously inhibited tumor growth and prolonged the overall survival of mice with tumor burden (Fig 7i, j).

Fig 7i, j: i. IVIS imaging of orthotopic tumor treated with FADS1 inhibitor, D5D-IN-326 (2mg/kg, p.o. for 2 weeks); j. Survival analysis of the mice burden with orthotopic tumor treated with D5D-IN-326.

17. Supplementary materials and methods: “Five million SW480 or HCT116 cells were injected into the left armpit of each null mouse.” Mice do not have armpits.

Response: Thanks for the reviewer’s good comments. We are so sorry about this inaccurate description and we injected CRC cells into the left anterior flank of each null mouse (showed in Supplementary materials: Xenograft model).

18. A schematic diagram should be shown that summarizes the results.

Response: Thanks for the reviewer’s good comments. We have drawn the schematic

diagram and showed in Fig 8.

Fig 8: FADS1-AA axis in CRC cells that modulate the intestinal microecology of enriched gram-negative bacteria via creating a high AA microenvironment. Enriched gram-negative microecology accelerated AA converting to PGE2 and eventually promotes CRC tumorigenesis.

19. The manuscript needs to be thoroughly proofread, as there are many typos and grammatical mistakes.

Response: Thanks for the reviewer's good comments. We have corrected the grammar mistakes and furnished up the revised manuscript.

Reference

1. Koundouros N, Karali E, Tripp A, Valle A, Inglese P, Perry NJS, et al. Metabolic Fingerprinting Links Oncogenic PIK3CA with Enhanced Arachidonic Acid-Derived Eicosanoids. *Cell*. 2020;181(7):1596-611 e27.
2. Tsoi H, Chu ESH, Zhang X, Sheng J, Nakatsu G, Ng SC, et al. Peptostreptococcus anaerobius Induces Intracellular Cholesterol Biosynthesis in Colon Cells to Induce Proliferation and Causes Dysplasia in Mice. *Gastroenterology*. 2017;152(6):1419-33 e5.
3. Wong SH, Zhao L, Zhang X, Nakatsu G, Han J, Xu W, et al. Gavage of Fecal Samples From Patients With Colorectal Cancer Promotes Intestinal Carcinogenesis in Germ-Free and Conventional Mice. *Gastroenterology*. 2017;153(6):1621-33 e6.
4. Tang L, Li J, Fu W, Wu W, Xu J. Suppression of FADS1 induces ROS generation, cell cycle arrest, and apoptosis in melanocytes: implications for vitiligo. *Aging*. 2019;11(24):11829-43.
5. Novichkova E, Chumin K, Eretz-Kdosha N, Boussiba S, Gopas J, Cohen G, et al. DGLA from the Microalga *Lobosphaera Incsa P127* Modulates Inflammatory Response, Inhibits iNOS Expression and Alleviates NO Secretion in RAW264.7 Murine Macrophages. *Nutrients*. 2020;12(9).
6. Qin XY, Lu J, Cai M, Kojima S. Arachidonic acid suppresses hepatic cell growth through ROS-mediated activation of transglutaminase. *FEBS open bio*. 2018;8(10):1703-10.
7. Borrelli A, Bonelli P, Tuccillo FM, Goldfine ID, Evans JL, Buonaguro FM, et al. Role of gut microbiota and oxidative stress in the progression of non-alcoholic fatty liver disease to hepatocarcinoma: Current and innovative therapeutic approaches. *Redox biology*. 2018;15:467-79.
8. Ballard JWO, Towarnicki SG. Mitochondria, the gut microbiome and ROS. *Cellular signalling*. 2020;75:109737.
9. Qiu Z, He Y, Ming H, Lei S, Leng Y, Xia ZY. Lipopolysaccharide (LPS) Aggravates High Glucose- and Hypoxia/Reoxygenation-Induced Injury through Activating ROS-Dependent NLRP3 Inflammasome-Mediated Pyroptosis in H9C2 Cardiomyocytes. *Journal of diabetes research*. 2019;2019:8151836.
10. Fu Y, Wang Y, Gao H, Li D, Jiang R, Ge L, et al. Associations among Dietary Omega-3 Polyunsaturated Fatty Acids, the Gut Microbiota, and Intestinal Immunity. *Mediators of inflammation*. 2021;2021:8879227.
11. Piazzini G, D'Argenio G, Prossomariti A, Lembo V, Mazzone G, Candela M, et al. Eicosapentaenoic acid free fatty acid prevents and suppresses colonic neoplasia in colitis-associated colorectal cancer acting on Notch signaling and gut microbiota. *International journal of cancer*. 2014;135(9):2004-13.
12. Cao SG, Chen R, Wang H, Lin LM, Xia XP. Cryptotanshinone inhibits prostaglandin E2 production and COX-2 expression via suppression of TLR4/NF-kappaB signaling pathway in LPS-stimulated Caco-2 cells. *Microbial pathogenesis*. 2018;116:313-7.
13. Higgins LG, Hayes JD. Mechanisms of induction of cytosolic and microsomal glutathione transferase (GST) genes by xenobiotics and pro-inflammatory agents. *Drug metabolism reviews*. 2011;43(2):92-137.
14. Zhang B, Jia WH, Matsuda K, Kweon SS, Matsuo K, Xiang YB, et al. Large-scale genetic study in East Asians identifies six new loci associated with colorectal cancer risk. *Nature genetics*. 2014;46(6):533-42.
15. Machate DJ, Figueiredo PS, Marcelino G, Guimaraes RCA, Hiane PA, Bogo D, et al. Fatty Acid Diets: Regulation of Gut Microbiota Composition and Obesity and Its Related Metabolic Dysbiosis. *International journal of molecular sciences*. 2020;21(11).

16. Chattopadhyay I, Dhar R, Pethusamy K, Seethy A, Srivastava T, Sah R, et al. Exploring the Role of Gut Microbiome in Colon Cancer. *Applied biochemistry and biotechnology*. 2021;193(6):1780-99.
17. Cournoux A, Dalmasso G, Martinez R, Buc E, Delmas J, Gibold L, et al. Bacterial genotoxin colibactin promotes colon tumour growth by inducing a senescence-associated secretory phenotype. *Gut*. 2014;63(12):1932-42.
18. D'Asheesh T A, Hussen BM, Al-Marzoqi AH, Ghasemian A. Assessment of oncogenic role of intestinal microbiota in colorectal cancer patients. *Journal of gastrointestinal cancer*. 2021;52(3):1016-21.
19. Sugimura N, Li Q, Chu ESH, Lau HCH, Fong W, Liu W, et al. *Lactobacillus gallinarum* modulates the gut microbiota and produces anti-cancer metabolites to protect against colorectal tumourigenesis. *Gut*. 2021.

REVIEWERS' COMMENTS

Reviewer #1 (Remarks to the Author):

Overall, I think the authors have significantly strengthened the manuscript with new data and have adequately addressed my comments.

One minor point would be that there is still some room for improvement in the abstract in terms of the language/grammar used.

Reviewer #3 (Remarks to the Author):

No more comments

Reviewer #4 (Remarks to the Author):

Authors replied to the most of the comments by original reviewers.

Comment/response about " Figure 2J and supplementary figure 2F are exactly the same figure (Tumor number per mice). Was the number of tumours identical in both experiments with the two different models or is this a mistake when compiling the figures?"

Response: Thank you very much for your careful review of our manuscript. Supplementary figure 2F was careless mistake and showed a repeat picture (repeat with Figure 2F, Fig 2i in revised Fig 2). We had revised it in Supplementary Fig 2g..."

remains not convincing. new Sup Fig 2g does not have N=6 as claimed

Point-By-Point responses:

Reviewer #1 (Remarks to the Author):

Overall, I think the authors have significantly strengthened the manuscript with new data and have adequately addressed my comments.

One minor point would be that there is still some room for improvement in the abstract in terms of the language/grammar used.

Response: Thank you very much for your careful review of our manuscript. We had modified the statements in the abstract carefully.

Reviewer #3 (Remarks to the Author):

No more comments

Reviewer #4 (Remarks to the Author):

Authors replied to the most of the comments by original reviewers.

Comment/response about " Figure 2J and supplementary figure 2F are exactly the same figure (Tumor number per mice). Was the number of tumours identical in both experiments with the two different models or is this a mistake when compiling the figures?"

Response: Thank you very much for your careful review of our manuscript. Supplementary figure 2F was careless mistake and showed a repeat picture (repeat with Figure 2F, Fig 2i in revised Fig 2). We had revised it in Supplementary Fig 2g..."

remains not convincing. new Sup Fig 2g does not have N=6 as claimed

Response: Thanks for the reviewer's valuable suggestion. I'm sorry again for causing this misunderstanding. We have claimed **n=6 mice per group** in the figure legend of Supplementary Fig 2g. And in Supplementary Fig 2g, each group had six plots. In NC group, the tumor number are 3, 4, 5, 6, 6, 6, so it looks like n=4, which showed in Source Data file.